# Alkyl Derivatives of Perylene Photosensitizing Antivirals: Towards Understanding the Influence of Lipophilicity

**DOI:** 10.3390/ijms242216483

**Published:** 2023-11-18

**Authors:** Igor E. Mikhnovets, Jiří Holoubek, Irina S. Panina, Jan Kotouček, Daniil A. Gvozdev, Stepan P. Chumakov, Maxim S. Krasilnikov, Mikhail Y. Zhitlov, Evgeny L. Gulyak, Alexey A. Chistov, Timofei D. Nikitin, Vladimir A. Korshun, Roman G. Efremov, Vera A. Alferova, Daniel Růžek, Luděk Eyer, Alexey V. Ustinov

**Affiliations:** 1Shemyakin-Ovchinnikov Institute of Bioorganic Chemistry, Miklukho-Maklaya 16/10, 117997 Moscow, Russia; mikhnovets.igor@gmail.com (I.E.M.); irinaspanina@gmail.com (I.S.P.); hathkul@gmail.com (S.P.C.); makrlist@gmail.com (M.S.K.); droplbox38@gmail.com (M.Y.Z.); evgeny.gulyak@gmail.com (E.L.G.); dobr14@yandex.ru (A.A.C.); timanikitin36@gmail.com (T.D.N.); v-korshun@yandex.ru (V.A.K.); efremov@nmr.ru (R.G.E.); alferovava@gmail.com (V.A.A.); 2Laboratory of Emerging Viral Diseases, Veterinary Research Institute, Hudcova 296/70, CZ-621 00 Brno, Czech Republicruzekd@paru.cas.cz (D.R.); ludek.eyer@vri.cz (L.E.); 3Institute of Parasitology, Biology Centre of the Czech Academy of Sciences, Branišovská 1160/31, CZ-370 05 České Budějovice, Czech Republic; 4Department of Experimental Biology, Faculty of Science, Masaryk University, CZ-625 00 Brno, Czech Republic; 5Department of Pharmacology and Toxicology, Veterinary Research Institute, Hudcova 296/70, CZ-621 00 Brno, Czech Republic; kotoucek@vri.cz; 6Department of Biology, Lomonosov Moscow State University, Leninskie Gory 1-12, 119234 Moscow, Russia; danil131054@mail.ru; 7Department of Chemistry, Lomonosov Moscow State University, Leninskie Gory 1-3, 119991 Moscow, Russia

**Keywords:** antivirals, perylene, photosensitizers, singlet oxygen, lipophilicity

## Abstract

Amphipathic perylene derivatives are broad-spectrum antivirals against enveloped viruses that act as fusion inhibitors in a light-dependent manner. The compounds target the lipid bilayer of the viral envelope using the lipophilic perylene moiety and photogenerating singlet oxygen, thereby causing damage to unsaturated lipids. Previous studies show that variation of the polar part of the molecule is important for antiviral activity. Here, we report modification of the lipophilic part of the molecule, perylene, by the introduction of 4-, 8-, and 12-carbon alkyls into position 9(10) of the perylene residue. Using Friedel–Crafts acylation and Wolff–Kishner reduction, three 3-acetyl-9(10)-alkylperylenes were synthesized from perylene and used to prepare 9 nucleoside and 12 non-nucleoside amphipathic derivatives. These compounds were characterized as fluorophores and singlet oxygen generators, as well as tested as antivirals against herpes virus-1 (HSV-1) and vesicular stomatitis virus (VSV), both known for causing superficial skin/mucosa lesions and thus serving as suitable candidates for photodynamic therapy. The results suggest that derivatives with a short alkyl chain (butyl) have strong antiviral activity, whereas the introduction of longer alkyl substituents (*n* = 8 and 12) to the perylenyethynyl scaffold results in a dramatic reduction of antiviral activity. This phenomenon is likely attributable to the increased lipophilicity of the compounds and their ability to form insoluble aggregates. Moreover, molecular dynamic studies revealed that alkylated perylene derivatives are predominately located closer to the middle of the bilayer compared to non-alkylated derivatives. The predicted probability of superficial positioning correlated with antiviral activity, suggesting that singlet oxygen generation is achieved in the subsurface layer of the membrane, where the perylene group is more accessible to dissolved oxygen.

## 1. Introduction

Enveloped viruses cause severe and frequently fatal diseases in humans and animals. Viruses from the *Herpesviridae* family are large, double-stranded DNA viruses with spherical enveloped virions that have been evolving alongside humans for more than 300 thousand years. They have developed numerous immunoevasive mechanisms to efficiently infect, replicate, and spread in human hosts [1]. Infections with members of the *Herpesviridae* family can manifest as a wide range of clinical symptoms, from lesions of the skin or mucous membranes to rubella and shingles, and sometimes can be associated with the development of severe encephalopathies or even tumor transformation. Although there are several approved drugs based on nucleoside analogs to treat herpesvirus infections, viral resistance and toxicity often limit their application [2]. Photodynamic therapy of superficial skin or mucosa lesions caused by some herpesviruses appears to be a promising alternative for the treatment of infections caused by herpes simplex virus-1 (HSV-1) and -2 or varicella zoster virus (VZV) [3,4,5]. Additionally, this therapy method is suitable for inactivating periodontal viruses, such as vesicular stomatitis virus (VSV), a member of the *Rhabdoviridae* family [6,7].

The outer envelope of virions is built upon a lipid bilayer acquired from a host cell in the course of virion maturation and egress. Intricate mechanisms of viral replication, such as lipid membrane fusion, play a crucial role in the propagation of enveloped virus infections [8]. The lipid membrane fusion occurs via pre-fusion → hemifusion → pore formation steps, initiating the infection process (Figure 1). Consequently, targeting viral membrane fusion has emerged as a promising strategy in the development of potent antiviral agents [9,10,11,12,13,14,15]. Virions lack molecular repair systems, so damage to their envelope by a direct-acting antiviral can inhibit fusion.

Amphipathic derivatives of pentacyclic aromatic hydrocarbon perylene are known as potent broad-spectrum antivirals against enveloped viruses [16,17,18,19,20]. Initially, their main mechanism of action was thought to be biophysical: the hydrophobic perylene moiety sticks to the lipid bilayer and acts as a wedge inhibiting the transition from positive to negative membrane curvature during the formation of a hemifusion intermediate [16,17]. However, further studies indicated that virus photosensitization through singlet oxygen photogeneration followed by unsaturated lipid peroxidation is the primary mechanism of action [21,22,23,24] (Figure 1). Although conjugated perylene chromophores act as the true pharmacophores responsible for the antiviral properties, the polar part of the molecule was shown to considerably modulate EC_50_ activity values [19,22]. Variations in the perylene moiety’s position have shown that 2- and 3-substituted perylene derivatives exhibit similar activities [24,25,26].

All previously reported perylene antivirals have been mono-substituted perylenes. Our aim was to introduce an additional lipophilic alkyl residue into the polycyclic aromatic moiety of perylene antivirals. This modification was expected to preserve both (1) the location of the perylene residue in the lipid bilayer upon action on an enveloped virion and (2) the spectral and photochemical/photosensitizing properties of the perylene chromophore. However, the additional alkyl could alter the location/orientation of the perylene moiety in the lipid bilayer, thereby modulating the efficacy of singlet oxygen action on its molecular target, unsaturated lipids [27,28]. Moreover, studies of activity in the absence of light might reveal a possible biophysical/mechanical component of the antiviral mechanism of action. It is possible that, for more lipophilic molecules, the hypothesis of intercalation into the membrane and inhibition of negative curvature formation in the viral membrane during fusion can be confirmed. Non-chromophore compounds that work presumably by this mechanism include surfactin [29,30,31,32,33] and other lipopeptides [34,35,36,37], persicamidines [38], and “molecular tweezers” [39,40]. Incorporating fatty acid moieties into drug molecules has proven to be a valuable strategy in the development of peptide- and protein-based drugs [41]. This approach can improve the pharmacokinetic properties of these drugs, enhancing their efficacy and safety. Examples of increased bioactivity resulting from lipophilic modification can be found among antioxidants [42,43,44] acting via protecting lipids from oxidative damage by ROS.

Therefore, we postulate that an alkyl substituent in the perylene residue, which can modulate the distance between the photosensitizer and the target while also enhancing its affinity for lipid membranes, could potentially alter the activity of perylene-based compounds against enveloped viruses.

## 2. Results and Discussion

### 2.1. Synthesis

Given that the proposed mechanism of action for perylene antivirals involves interactions with the viral lipid membrane [22,23], it is a logical step to modulate the lipophilicity of these compounds by introducing alkyl residues into the perylene portion. Our objective was to synthesize a series of compounds with increased lipophilicity of the perylene moiety. Specifically, we aimed to obtain derivatives with alkyl tails containing 4, 8, and 12 carbons in the hydrocarbon chain, representing an incremental change in the lipophilicity of the molecules by 2.1 ClogP, resulting in calculated ClogP values ranging from about 4 to 15. Additionally, we selected hydrophilic moieties for these compounds to match those used in previously synthesized compounds in our lab that lacked alkylation of the perylene core. This allowed us to expand the chemical diversity of compounds for antiviral experiments and structure–activity relationship (SAR) analysis.

The synthesis of perylenylethynyl nucleoside compounds was performed in two parts: in the first part, we synthesized 9(10)-alkyl derivatives of 3-ethynyl perylenes **5a–c** in five steps (Figure 1), and, in the second part, we conducted a synthesis of target compounds by Sonogashira reaction with compounds **5a–c**, followed by deprotection.

3-Acylperylenes **1a–c** were obtained from the starting perylene using a Friedel–Crafts reaction [45,46,47]. Then, their reduction was performed using the Huang–Minlon modification of the Wolf–Kishner reaction [48], affording 3-alkylperylenes **2a–c**. The reaction products were isolated through extraction with hexane in a Soxhlet extractor. It is worth noting that the yield of 3-butanoylperylene (**1a**) turned out to be less than that of the other two compounds, **2b** and **2c**, due to their higher solubility in hexane, resulting in more efficient extraction. The second Friedel–Crafts acylation reaction was carried out similarly to the first stage, following reported conditions [46]. During this step, a mixture of two isomeric compounds **3a–c** (3,9- and 3,10-acetylalkylperylenes) was formed (Figure 1), as evidenced by NMR spectra. Attempts to separate this mixture using column chromatography were unsuccessful. However, since these compounds target the lipid bilayer, using a mixture of two structural isomers rather than an individual substance is unlikely to significantly affect their antiviral activity. Compounds **5a–c** were obtained in two steps from acetylalkylperylenes **3a–c** by Vilsmeer–Haack–Arnold formylation, followed by Bodendorf fragmentation [49,50], as described in [46]. The ratio of 3,9- and 3,10-alkylethynylperylenes, and therefore all other compounds, is determined from ^1^H NMR spectra and is 34% to 66% for 3,10- to 3,9-alkylethynylperylenes (Appendix A).

In the second part of the synthesis, we performed Sonogashira coupling of 3′,5′-*O*-acyl protected 5-iodouracil nucleosides with 3-ethynyl-9(10)-alkylperylenes **5a–c** followed by deprotection with K_2_CO_3_ in a MeOH/H_2_O/DCM mixture similarly to previously reported procedures [16,25] (Figure 2). Propionyl, an acyl-type protecting group, allows easier purification of protected intermediates **6a–c**, **7a–c**, and **8a–c** by column chromatography, in contrast to the use of silyl-type protecting groups, e.g., TBDMS; this is consistent with a recent observation [51]. The desired nucleoside compounds **9a–c**, **10a–c**, and **11a–c** were isolated with reasonable yields.

Very recently, we reported the antiviral properties of perylenylethynylphenols [24]. To prepare their alkylated analogs **12a–c**, **13a–c**, and **14a–c**, we also employed direct Sonogashira coupling of alkylethynylperylenes **5a–c** with the corresponding iodophenols (Figure 3). It is noteworthy that a heavy halogen atom in the dye (bromine in compounds **14a–c**) is believed to enhance the quantum yield of singlet oxygen photogeneration [52,53]; therefore, these compounds were of particular interest due to the potential influence of the bromine atom on both the quantum yield of singlet oxygen generation and antiviral activity.

Then, we prepared alkylperylenylthiophenecarboxylic derivatives **15a–c** (Figure 4). The parent 5-(perylene-3-yl)thiophenecarboxylic acid is known to be a potent antiviral and a good singlet oxygen photogenerator [20,23]. To prepare its alkyl derivatives, we conducted a reaction of ethyl thioglycolate with *E*/*Z*-3-chloro-3-(9(10)-alkylperylen-3-yl)acroleins **4a–c** in the presence of KOH. The procedure was similar to a previously reported one [20].

### 2.2. Spectral Properties, Solubility and ^1^O_2_ Photogeneration

The absorption spectra of the synthesized compounds based on alkylperylenes exhibit similarities within their respective groups: nucleoside derivatives (**9**–**11a**–**c**), phenolic derivatives (**12**–**14a**–**c**), and thiophenecarboxylic derivatives (**15a–c**) (Figure 1).

It should be noted that the absorption spectra of *arabino*- and *deoxy*-uridine derivatives with lauroylperylene (**9c** and **11c**) exhibit a different shape compared to other nucleoside derivatives. In polar solvents (methanol, DMSO, 1:1 mixture of chloroform and methanol), these compounds are likely to form aggregates, as indicated by the absorption spectra with broadened peaks (Figure 2). This phenomenon can be explained by the ratio of hydrophilic and hydrophobic parts of molecules and a certain conformation (*arabino*- and *deoxy*-) of the sugar in these molecules, which is favorable for the formation of aggregates, similar to one demonstrated earlier [54]. The solvent also plays an important role in the aggregation of dye molecules, and often more polar solvents, particularly those containing a hydroxyl group, enable the aggregation of hydrophobic molecules [55].

Fluorescence and excitation spectra of the synthesized compounds based on alkylperylenes were shown to be similar for the groups: nucleoside derivatives (**9–11a–c**), phenolic derivatives (**12–14a–c**), and thiophenecarboxylic derivatives (**15a–c**). As in the case of absorption spectra, compounds **9c** and **11c** in methanol solution existed in an aggregated state, resulting in a broadening of both the fluorescence and excitation spectrum peaks for these compounds (Figure 3).

The fluorescence properties of the compounds are summarized in Table 1. Interestingly, compounds derived from *para*-phenol **12a–c** show a Stokes shift 3 nm larger than compounds derived from *meta*-phenol **13a–c** and *ortho*-bromo, para-phenol **14a–c**. Also, compounds of thiophenecarboxylic acid derivatives **15a–c** exhibit a Stokes shift of 75 nm, larger than for other compounds from the series, which may indicate higher mobility of the thiophene moiety relative to perylene compared to compounds containing a triple bond [56]. It is important to note that the formation of aggregates occurs even in solutions that are much more dilute than in the absorption measurement experiment, as evident from the shape of the spectra (Figure 3).

To assess the usability of the synthesized compounds in cell media, it is important to determine their solubility in these media. Perylene-based antiviral compounds are poorly soluble in water, making it difficult to measure their solubility in pure water. Therefore, in order to evaluate the solubility of the resulting compounds **9–15a–c**, we measured the amount of compound that could be solubilized with 15% DMSO in water. Given that these compounds are amphiphilic molecules with both hydrophobic and hydrophilic parts, they most likely do not form an ideal solution in 15% DMSO in water, but rather exist in a solubilized micelle form. To perform this measurement, the compound was dissolved in DMSO and then diluted with water to achieve a solution containing 15% DMSO. The solution was centrifuged, and the resulting supernatant was lyophilized (evaporated) to dryness. Subsequently, the dry residue was dissolved in DMSO, and the concentration of the compound dissolved in DMSO was determined spectrophotometrically. Using these parameters, the solubility of each compound in 15% DMSO in water was calculated (Table 1).

*Deoxy*- and *arabino*-nucleoside compounds with a butyl tail (**9a**, **11a**) show relatively high solubility in 15% DMSO in water. In contrast, the *ribo*- derivative with a butyl tail (**10a**) showed lower solubility. This difference can be attributed to the positioning of hydroxyl groups in the sugar moiety, which could affect the formation of solubilized micelles. Nucleoside compounds with an octyl tail (**9b**, **10b**, **11b**) have comparable solubility to those with a butyl tail (**9a**, **11a**). However, the compound based on arabinonucleoside (**9b**) dissolves approximately 1.5 times less effectively than its *deoxy*- and *ribo*- analogs (**10b**, **11b**). Nucleoside compounds with a dodecyl tail (**9c**, **10c**, **11c**) demonstrated lower solubility relative to compounds with a butyl and octyl tail, which is associated with an increase in the length of the alkyl tail and, accordingly, changes in the distribution of hydrophobic and hydrophilic parts of the molecule, which leads to a change (reduction) in the micellization abilityof.

Phenol-based compounds are inherently more hydrophobic than nucleoside-based compounds. As expected, this higher hydrophobicity reduces their solubility in polar solvents, as observed in the measurements of solubilization in 15% DMSO in water. For derivatives of alkylperylenes based on *meta*-phenol (**13a–c**), there is an increase in solubility with the elongation of the alkyl chain. This can be explained by a better ability to form micelles. Compounds based on *para*-phenol (**12a–c**) exhibit a solubility pattern that reaches a maximum for the compound with an octyl alkyl tail (**12b**). On the other hand, compounds based on *ortho*-bromo and *para*-phenol (**13a–c**) are the most hydrophobic among those synthesized. Consequently, they display low solubility, comparable to that of thiophenecarboxylic acid derivatives (**15a–c**).

When irradiated with light, the photosensitizer absorbs light energy and converts dissolved oxygen into singlet oxygen form, ^1^O_2_. To quantify the singlet oxygen generation rate, a singlet oxygen trap, DPBF (1,3-diphenylisobenzofuran), reacts with singlet oxygen, leading to changes in its spectral properties. Specifically, the absorption spectrum of the trap changes, and it ceases to absorb light in a certain wavelength range. The rate at which absorption decreases in this range is proportional to the singlet oxygen generation rate. By comparing the absorption reduction rates of the trap in the initial linear section obtained from experiments with antiviral compounds based on perylene and riboflavin, we determined the relative quantum yield of singlet oxygen generation for the compounds. The quantum yield values of the compounds are provided in Table 1. However, it is important to note that the singlet oxygen trap is not photostable and undergoes “bleaching” upon irradiation, which can introduce some errors into the measurement results.

The data in Table 1 reveals that, on average, compounds based on alkylperylenes show lower singlet oxygen generation yields compared to their analogs without an alkyl tail (Figure 4). Moreover, compounds with *para*-hydroxy-*meta*-bromophenyl and *para*-hydroxyphenyl hydrophilic parts (**12a–c**, **14a–c**) generate singlet oxygen with a higher quantum yield than compounds with a nucleoside hydrophilic part (**9–11a–c**). In contrast, compounds with a *meta*-hydroxyphenyl hydrophilic moiety (**13a–c**) show lower quantum yields than other phenolic derivatives, similar to nucleoside derivatives of alkylperylenes. A similar trend is observed for compounds without alkyl modification. *Para*-phenolic (**12a–c**) and *meta*-phenolic (**13a–c**) derivatives have comparable quantum yields, which are higher than those for nucleosides (**11a–c**, **9a–c**, **10a–c**) and bromophenolic (**14a–c**) derivatives. Among the compounds tested, thiophenecarboxylic derivatives **15** and **15a–c** exhibited the most potent singlet oxygen generation rate.

Such a difference in quantum yields between alkyl derivatives and “control” compounds without alkyl can be attributed to aggregation, where lipophilic molecules tend to stick together in a polar solvent. A similar phenomenon has been observed for porphyrins; as the concentration of the photosensitizer increases, the quantum yield of singlet oxygen decreases [57,58]. Also, for methylene blue, a relatively hydrophobic photosensitizer, a decrease in the generation of singlet oxygen with increasing polarity of the solvent mixture was shown [59]. The replacement of the nucleoside moiety by the phenol group results in an increased quantum yield of singlet oxygen generation. Thus, it was found that phenolic derivatives generate singlet oxygen approximately 1.5 times better than nucleoside derivatives, and alkyl modification tends to diminish the photosensitization activity of the compounds.

### 2.3. Molecular Dynamics Studies

In order to investigate the behavior of perylene derivatives **12** and **14** with alkyl chains of different lengths—C_0_ (**12**,**14**), C_4_ (**12a**,**14a**), C_8_ (**12b**,**14b**), and C_12_ (**12c**,**14c**)—in a lipid bilayer, we performed a set of molecular dynamics (MD) simulations. In MD trajectories, the initially water-exposed perylene derivatives are rapidly (within 20–50 ns) inserted into the lipid bilayer, regardless of alkyl tail length. In the membrane-bound state, the plane of the perylene group was perpendicular to that of the bilayer, with an MD-averaged value of the angle between these planes being 91 ± 18°. Due to the rotational symmetry and planarity of the compounds, to define their orientation in the bilayer, we only calculated the evolution of the tilt angle subtended by the molecular long axes with respect to the bilayer normal (Appendix A). The embedded molecules adopted two major orientations relative to the membrane, with a tilt angle of approximately 0°, corresponding to the parallel position (along the lipids’ acyl chains, **state 1**), and 90° (along the lipids’ phosphate groups, **state 2**) (Figure 5A). The alkyl tail was found to stabilize the parallel position: state 2 was more frequently encountered in C_0_ derivatives (Figure 5B). Membrane penetration depths of all molecules in state 1 (tilt angle < 60°) were almost equal (Z~0.6 nm), whereas in state 2 (tilt angle ≥ 60°) C_0_-compounds “float” closer to the membrane surface (Figure 5C and Appendix A). Moreover, solvent accessibility of the perylene group varied significantly between states: the probability of contact (distance < 0.6 nm) with more than one H_2_O molecule was 27% and 84% in states 1 and 2, respectively (Appendix A). Thus, we hypothesize that photosensitization of the compounds is associated with their localization in the subsurface layer of the membrane when the perylene group is more accessible to the solvent and dissolved O_2_ molecules.

### 2.4. Antiviral Properties

We measured the antiviral activity of synthesized and control compounds against VSV and HSV-1. These viruses were chosen because of a light-dependent manner of action of compounds based on perylene; therefore, only superficial viruses could be treated in a real-world scenario. Initial assessment of antiviral activity against VSV demonstrated that, generally, lipophilized compounds showed less potent antiviral activity than their analogs without lipophilization (Table 2). Cytotoxicity of the compounds towards HER-293T cells (Appendix A) and VSV TCID_50_ (Appendix A) reduction are presented in the Appendix A.

Deep investigations of the antiviral properties of the compounds were performed with HSV-1. With regard to anti-HSV-1 studies, we first examined the cytotoxicity of alkylated perylene compounds in Vero cells, as this cell line is commonly used to culture HSV-1 and for anti-HSV-1 assays (Figure 6A). Most of the alkylated perylene compounds tested (at 10 µM) were not cytotoxic to Vero cells after a 48 h incubation. In general, compounds **10a**, **12a,** and **14b** decreased cell viability to ~80% compared to controls. In contrast, compound **11b** increased the intensity of cellular metabolism, as evidenced by cell viability of ~120%. Control compounds **9**–**15** showed no cytotoxicity in Vero cells; perylenylethynylphenols **12**, **13,** and **14** slightly increased cell viability (to 109–122% compared to DMSO-treated cells) (Figure 6A), as previously described [24].

We then performed a general screening of anti-HSV-1 activities in Vero cells; cells were infected with HSV-1 (strain MacIntyre) and incubated for 48 h with the tested compounds (10 µM). Compounds with short alkyl substituents (*n* = 4) attached to the perylene core appeared to be more active against HSV-1 than those with longer alkyl chains (*n* = 8 or 12) (Figure 6B). Among the alkylated nucleoside-based perylene compounds, the 9(10)-butyl-substituted perylenylethynyl nucleosides **9a**, **10a,** and **11a** exhibited potent anti-HSV-1 activity and completely inhibited viral replication in Vero cells, regardless of ribose configuration (*arabino*-, *ribo*-, or *deoxy*-, respectively). Interestingly, the 9(10)-octyl-substituted perylenylethynyl nucleosides **9b**, **10b**, and **11b** were only moderately active (a reduction in viral titer of one and two orders of magnitude, respectively). The 9(10)-dodecyl-substituted perylenylethynyl nucleosides **9c** and **10c** were completely inactive against HSV-1 (Figure 6B).

Similar structure–activity relationships were observed for alkyl-substituted perylenylethynylphenols (Figure 6B). The only two compounds from this series that showed anti-HSV-1 activity were **13a** and **12a**, both of which have a 9(10)-butyl substitution of the perylenylethynyl core. Compounds with longer alkyl chains (*n* = 8 and 12) were found to be completely inactive against HSV-1. We did not observe any differences in anti-HSV-1 activity of perylenylethynylphenols bearing a hydroxyl group in either the *meta*- or *para*-position of the ethynylperylene moiety. Interestingly, the presence of a heavy halogen atom did not improve the anti-HSV-1 activity of the perylenylethynylphenols studied; only the 9(10)-butylated derivative **14a** had a slight reduction in HSV-1 titer (about one order of magnitude compared with controls) (Figure 6B).

Finally, alkylperylenylthiophene carboxylic acids **15a** and **15b** caused complete inhibition of HSV-1, whereas **15c** showed no anti-HSV-1 activity (Figure 6B). Compound **15b** was the only 9(10)-octyl-substituted perylene compound from the tested series with strong anti-HSV-1 activity. The superior antiviral activity of the alkylperylenylthiophene carboxylic acids is likely related to their increased singlet oxygen photogeneration, which was 3 to 4 fold higher compared with other alkylated perylene compounds tested (Table 1). Increased singlet oxygen photogeneration ability was also recently observed for numerous non-alkylated perylenylthiophene compounds [23].

The alkylated perylene compounds **9a**, **10a**, **11a**, **12a**, **13a**, **15a**, and **15b** showed higher antiviral potency compared to acyclovir, which completely inhibited HSV-1 replication at 50 µM; at 10 µM, acyclovir caused only a moderate anti-HSV-1 effect (2–3 orders of magnitude decrease in viral titer) (Figure 6B). As expected, non-alkylated perylene derivatives (control compounds) showed strong anti-HSV-1 activity (Figure 6B), which is consistent with our previous studies [23,24]. As previously demonstrated in other perylene compounds [23], derivatives **9a**, **10a**, **11a**, **12a**, **13a**, **15a**, and **15b** (at 10 µM) inactivated HSV-1 virions only in supernatant medium (Figure 6B), but did not suppress viral replication and spread in HSV-1-infected Vero cell culture (Figure 6C).

It can be concluded that the length of the alkyl chain attached to the perylene core determines the anti-HSV-1 activity of the alkylated perylenylethynyl derivatives. Moreover, the antiviral activity is closely related to the hydrophobicity (and solubility; see Table 1) of the compounds, and the formation of insoluble aggregates in derivatives with a long alkyl chain (mostly dodecyl) strongly affects the anti-HSV-1 activity of the compounds. As previously demonstrated in other perylene compounds [23], derivatives **9a**, **10a**, **11a**, **12a**, **13a**, **15a**, and **15b** (at 10 µM) inactivated HSV-1 virions only in supernatant medium (Figure 6B), but did not suppress viral replication and spread in HSV-1-infected Vero cell culture (Figure 6C).

### 2.5. Light-Induced HSV-1 Inactivation Effect

Based on our photochemical data, alkylated perylene derivatives are expected to act as photosensitizers that directly inactivate HSV-1 particles via the singlet oxygen-photogeneration mechanism. We therefore investigated the virus-inactivating effect of alkylated perylene derivatives after blue light irradiation; a mixture of HSV-1 (titer of 10^4^ PFU/mL) and selected compounds (0–10 µM) was irradiated with blue light (465–480 nm at an approximate power density of 30 mW/cm^2^) for 10 min, and virus viability was determined by plaque assay (Figure 7D). Virus incubated with the compounds in daylight for 10 min served as control (non-irradiated) samples (Figure 7A).

HSV-1 treated with compounds **9a**, **10a**, **11a**, **12a**, **13a**, **15a**, and **15b** (all of which showed anti-HSV-1 activity in the previous antiviral assay) and irradiated with blue light showed a sharp decrease in viability compared with non-irradiated virus samples (incubated in daylight) (Figure 7B,C,E,F, Table 3). This effect was most pronounced for compounds **9a**, **10a**, and **11a** (EC_50_ values decreased 21.9, 20.3, and 18.6 fold, respectively, compared with non-irradiated virus samples). For compounds **12a** and **13a**, the virus-inactivating effect under blue light was somewhat weaker, with EC_50_ values decreasing only 3 to 4 fold). Compound **15a** showed strong virus-inactivating activity after blue light irradiation, which was comparable to that of compounds **9a**–**11a** (EC_50_ of around 0.1 µM for all compounds). Extremely strong virus inactivation activity of **15a** was observed even after incubation of the virus–compound mixture in daylight (EC_50_ of 0.61 µM). In contrast, compound **15b** showed no virus-inactivating effect in daylight (EC_50_ > 10 µM), but, after blue light irradiation, the EC_50_ value decreased >3.5 fold (Figure 7B,C,E,F, Table 3).

We further compared the light-dependent HSV-1-inactivating activity of selected alkylated perylene compounds (**9a**, **10a**, **11a**, **12a**, **13a**, **15a,** and **15b**) with their non-alkylated counterparts (control compounds). After blue light irradiation, alkylated perylenylethynyl nucleosides **9a**–**11a** did not differ in their HSV-1-inactivating activity from the non-alkylated derivative **10**. However, alkylated perylenylethynylphenols **12a** and **13a** and perylenylthiophene carboxylic acids **15a** and **15b** were almost 6 fold and 3 fold, respectively, less active than their non-alkylated counterparts **12** and **15** (Figure 7E,F, Table 3).

Interestingly, irradiation with blue light strongly increased the anti-HSV-1 activity of compounds **9b** and **10b** (10 µM) and resulted in complete inactivation of HSV-1. A moderate light-dependent virus inactivation effect was observed for compounds **9c**, **12c**, and **14a** (a decrease in virus titer of 1 to 1.5 orders of magnitude compared with controls). In contrast, no virus-inactivating effect was observed after irradiation of HSV-1 treated with **12b**, **13b**, **13c**, **14b**, **14c**, and **15c** (Figure 7G,I).

From our structure–activity relationship study, the following conclusions can be drawn: (i) alkylated perylenylethynyl nucleosides show the highest solubility in 15% DMSO of the entire series of compounds tested, but they exhibit the lowest singlet oxygen photogeneration ability. Their 9(10)-butylated derivatives show strong HSV-1-inactivation activity; moderate antiviral activity was also observed with the 9(10)-octylated analogues. (ii) Alkylated perylenylethynylphenols show low solubility but higher singlet oxygen photogeneration capacity compared to nucleoside analogues. Their 9(10)-butylated derivatives are active against HSV-1, but their HSV-1-inactivation activity is approximately 30 fold lower than that of nucleosides. Other perylenylethynylphenols, including brominated derivatives, show no HSV-1-inactivation activity. (iii) Alkylperylenylthiophene carboxylic acids exhibit similar low solubility to perylenylethynylphenols, but have the most pronounced singlet oxygen photogeneration. This is apparently reflected in their superior HSV-1-inactivation activity; these compounds showed the highest antiviral activity of the entire series tested (octylated derivatives are also active). (iv) Most of 9(10)-ocylated or dodecylated derivatives show no activity against HSV-1 and exert lower solubility and lower singlet oxygen generation capacity than derivatives with the 9(10)-butyl substitution. (v) Non-alkyated perylene compounds were either equally active against HSV-1 (nucleosides) or showed higher HSV-1-inactivation potency than their 9(10)-butylated counterparts (phenols and thiophene-containing derivatives), particularly when irradiated with blue light. They show significantly higher singlet oxygen photogeneration capacity compared with alkylated perylene compounds.

Light-induced virus inactivation activity based on singlet oxygen photogeneration is not an exclusive property of perylene derivatives, but has been demonstrated for a number of chemicals that have the potential to be used as photosensitizers for photodynamic antiviral (or antitumor) therapy. These include, for example, methylene blue [60,61], arylidene rhodanines [11,62,63,64,65], BODIPY [66] porphyrins and porphyrin-like compounds [67,68,69], riboflavin [70] and thiopyrylium [71] derivatives, and many others.

### 2.6. Light-Dependent HSV-1-Inactivation Activity

Similar to other recently described perylene compounds [21,23,24], the alkylated perylene derivatives lost their HSV-1-inactivating activity when the entire experiment, including sample preparation and plaque assays, was performed under red light (624 ± 20 nm, a wavelength far from the excitation/absorption maxima of the compounds studied). Under these experimental conditions, the compounds showed no HSV-1-inactivating activity at either concentration tested, 10 and 100 µM. We also intentionally used a high concentration of the compounds (100 µM) to demonstrate that the compounds had no activity in the absence of excitation light, even at concentrations 10 fold higher than those used in conventional antiviral assays (Figure 7H,J). Our results deny the hypothesis that the mechanism of the antiviral effect of perylene compounds could be based on the geometry of the molecule (a biophysical or intercalation mechanism) [25]. On the contrary, it seems that the antiviral activity of these compounds is exclusively light dependent and based on the photogeneration of singlet oxygen with antimicrobial properties, as previously described [21].

### 2.7. Photocytotoxicity

Because of the potency of the alkylated perylene derivatives to be used as photosensitizing antiviral agents, we also investigated light-induced cytotoxicity (photocytotoxicity) of selected compounds **9a**, **9b**, and **9c** (Figure 7K)**.** Irradiation of Vero cells treated with the compounds with blue light for 10 min did not increase the cytotoxicity of the tested compounds compared with Vero cells incubated with the compounds in daylight. In contrast, irradiation actually resulted in a slight increase in cell viability (~115–128%) of cells treated with the compounds (Figure 7L). Similar results were recently described for non-alkylated perylenylethynylphenols used for photocytotoxicity studies with *Felis catus* kidney cortex (CRFK) cells [24]. In contrast, some perylenylethynyl derivatives were highly cytotoxic to Vero cells after irradiation with blue light [23]. It is obvious that the photocytotoxicity of perylene compounds depends on their ability to generate singlet oxygen, but it can also be strongly influenced by the type and biological properties (metabolic activity, proliferation ability, etc.) of the cell line tested.

### 2.8. Mechanistic Studies

The HSV-1-inactivation studies described above (Figure 7A–J) were based on treating the virus with the compounds and then estimating the viability of the virus. These studies suggest that the alkylated perylenes act directly on HSV-1 particles and reduce/eliminate HSV-1 infectivity. To further elucidate the mechanisms of antiviral efficacy of alkylated perylene derivatives, we used specific cell-based assays (Figure 8A,B) to demonstrate that these compounds interact with the HSV-1 envelope and suppress the process of virus–cell fusion.

The interaction of alkylated perylene compounds with the HSV-1 envelope was demonstrated using a cell-based intercalation assay (Figure 8A). HSV-1 was pre-incubated with compounds **9a** and **11a** (at 0, 0.08, 2, and 10 µM) at 37 °C. Vero cells were then infected at 4 °C, the non-adsorbed virus was washed off, and the virus-infected cells were cultured for an additional 5 days. The results showed that pretreatment of HSV-1 with the tested compounds (at 2 and 10 µM) completely inhibited the ability of the virus to form plaques on the Vero cell monolayer (Figure 8C,E). These results suggest that alkylated perylene compounds interact with the HSV-1 envelope. We can speculate that the amphipathic perylene compounds intercalate between the (phospho)lipids of the viral envelope membrane, as later demonstrated using liposome-based studies.

Using a cell-based fusion assay, we also demonstrated that alkylated perylene derivatives block the virus–cell fusion process (Figure 8B). In the fusion assay, HSV-1 was attached to Vero cells at 4 °C, but virus–cell fusion was blocked due to the low temperature. After washing off the non-adsorbed virus and adding the tested compounds **9a** and **11a** (0, 0.08, 2, and 10 µM), the temperature was suddenly increased to 37 °C to allow fusion to occur. Subsequently, the infected cells were cultured for another 5 days to observe a reduction in plaque numbers in infected cells treated with the two tested compounds (Figure 8B). The effect was most pronounced at 2 and 10 µM, while at a concentration of 0.08 µM, plaque numbers were similar to controls. In addition to reduced plaque numbers, HSV-1 formed significantly smaller plaques in Vero monolayers in the presence of **9a** and **11a** (Figure 8D,F). Our results suggest that alkylated perylene compounds block the virus–cell fusion machinery.

The mechanism of action of perylene-based compounds has been extensively studied in non-alkylated perylene derivatives [19,20,21,22,23,24,25], and the results indicate that these compounds inhibit the formation of negative membrane curvature during virus–cell fusion, which eventually leads to the blockage of the virus–cell fusion process. Based on the structural similarities between the previously described compounds [19,20,21,22,23,24,25] and the alkylated perylene derivatives used in this work, we can hypothesize the same mechanism of their antiviral activity.

### 2.9. Liposome-Based Studies

Furthermore, we investigated whether alkylated perylene compounds were incorporated into liposomes representing a protein-free lipid membrane model system with a defined lipid composition. For this experiment, we chose the alkylated perylenylethynyl-arabino-nucleosides **9a**, **9b**, and **9c** (10 µM), for which we observed decreasing anti-HSV-1 efficacy depending on the length of the alkyl chain attached to the perylene core, with **9a** being the most active compound (Figure 6B). We compared their fluorescence spectra with those of control compound **9**, which is a non-alkylated perylenylethynyl-arabino-nucleoside derivative.

All compounds tested gave very low fluorescence signals when dissolved in PBS (Figure 9A–D, blue dashed lines). However, after the addition of unilamellar liposomes (EPC/Chol of 70/30 mol%), the compounds showed significant fluorescence activity with two emission maxima, whose position in the fluorescence spectrum differed slightly depending on the identity of the compound (Figure 9A–D, red lines). These changes in fluorescence properties indicated that the compounds formed poorly fluorescent micelles or insoluble aggregates in PBS, whereas they were rapidly incorporated into liposome membranes and converted to a membrane-bound, highly fluorescent form. These changes were observed for all three structural forms **9a**–**9c** regardless of the length of their 9(10)-alkyl substituent. Moreover, a very similar behavior was observed for their non-alkylated counterpart **9** (Figure 9A–D). Our results show that both the alkylated and non-alkylated perylene derivatives have a strong affinity for lipid bilayers, which confirms our hypothesis that these compounds are likely to be also incorporated into viral membrane envelopes. Although all tested photosensitizers interacted with our liposome model with approximately the same intensity (as evident from the spectra obtained, Figure 9A–D), we hypothesize that the differences in their virus-inactivating activity were likely due to (i) lower solubility of derivatives with the long alkyl (Table 1) and/or (ii) their location in the lipid membrane bilayer (**state 1** in Figure 5), which may affect the contact of the perylene group with dissolved oxygen molecules.

## 3. Materials and Methods

Reagents and solvents were purchased from commercial suppliers and used as received. All solvents were purified according to standard procedures [72]. Analytical thin-layer chromatography was performed on Kieselgel 60 F_254_ precoated aluminum plates (Merck, Darmstadt, Germany). Silica gel column chromatography was performed using Merck Kieselgel 60 0.040–0.063 mm; 700 MHz ^1^H and 176 MHz ^13^C NMR spectra were recorded on a Bruker instrument and calibrated using residual solvent as the internal reference (for DMSO-*d*_6_—2.50 ppm for ^1^H and 39.52 ppm for ^13^C; for CDCl_3_—7.26 ppm for ^1^H and 77.16 ppm for ^13^C). ^1^H NMR coupling constants are reported in hertz (Hz) and refer to apparent multiplicities. NMR spectra of the compounds are provided in Appendix A. High-resolution mass spectra (HRMS) of compounds were recorded on a Thermo Scientific Orbitrap LTQ XL mass spectrometer with electrospray ionization (ESI). UV spectra were recorded on a Varian Cary 100 spectrophotometer. Fluorescence spectra were recorded on a PerkinElmer LS55 luminescence spectrometer.


**General procedure for the synthesis of 3-acylperylenes (1a–c):**


Acid chloride of the corresponding acid (1.2 equiv) was added to perylene (1 equiv) dissolved in chlorobenzene (10 g/L), after which a solution of AlCl_3_ (1.3 equiv) in nitromethane (0.1 g/mL) was added dropwise using an addition funnel. The progress of the reaction was monitored by TLC. After the complete conversion of perylene, the mixture was poured into a 10% solution of HCl in water, stirred for 30 min, and then the product was extracted with 3 portions of CH_2_Cl_2_. The organic phase was dried over anhydrous sodium sulfate and evaporated to dryness under vacuum. Pure substances **1a–c** in the form of an orange powder were isolated by column chromatography (eluent: chlorobenzene).

**3-Butanoylperylene (1a)** was prepared from 10.0 g (39.6 mmol) of perylen; yield 12.3 g (95%). Orange solid. ^1^H NMR (CDCl_3_, δ, ppm, J/Hz): δ 8.47 (d, *J =* 8.4 Hz, 1H), 8.15–8.06 (m, 3H), 8.00 (d, *J =* 7.8 Hz, 1H), 7.73 (d, *J =* 7.8 Hz, 1H), 7.67 (d, *J =* 8.0 Hz, 1H), 7.63 (d, *J =* 8.0 Hz, 1H), 7.54–7.48 (m, 1H), 7.42 (td, *J =* 7.7, 4.4 Hz, 2H), 2.99 (t, *J =* 7.3 Hz, 2H), 1.84 (h, *J =* 7.4 Hz, 2H), 1.06 (t, *J =* 7.4 Hz, 3H). ^13^C NMR (CDCl_3_, δ, ppm): δ 204.3, 135.2, 135.0, 134.5, 131.9, 131.2, 131.0, 130.3, 129.3, 129.1, 128.3, 128.2, 128.1, 128.1, 126.8, 126.6, 125.8, 121.6, 121.0, 120.8, 118.8, 44.0, 18.4, 14.0.

**3-Octanoylperylene (1b)** was prepared from 10.0 g (39.6 mmol) of perylene; yield 12.2 g (81%). Orange solid. ^1^H NMR (CDCl_3_, δ, ppm, J/Hz): δ 8.47 (dd, *J =* 8.5, 1.0 Hz, 1H), 8.20–8.13 (m, 3H), 8.07 (d, *J =* 7.9 Hz, 1H), 7.77 (d, *J =* 7.9 Hz, 1H), 7.71 (d, *J =* 8.0 Hz, 1H), 7.67 (d, *J =* 8.0 Hz, 1H), 7.54 (dd, *J =* 8.5, 7.4 Hz, 1H), 7.46 (td, *J =* 7.7, 1.1 Hz, 2H), 3.02 (t, *J =* 7.4 Hz, 2H), 1.80 (*p*, *J =* 7.5 Hz, 2H), 1.42 (dddd, *J =* 10.6, 8.8, 6.9, 1.5 Hz, 2H), 1.39–1.34 (m, 2H), 1.34–1.28 (m, 4H), 0.90 (t, *J =* 6.9 Hz, 3H). ^13^C NMR (CDCl_3_, δ, ppm): δ 204.5, 135.1, 134.9, 134.5, 131.9, 131.2, 131.0, 130.3, 129.3, 129.1, 128.3, 128.2, 128.1, 128.1, 126.8, 126.6, 125.8, 121.6, 121.0, 120.8, 118.8, 42.1, 31.8, 29.5, 29.2, 25.0, 22.7, 14.1.

**3-Dodecanoylperylene (1c)** was prepared from 10.0 g (39.6 mmol) of perylene; yield 12.0 g (70%). Orange solid. ^1^H NMR (CDCl_3_, δ, ppm, J/Hz): δ 8.48 (d, *J =* 8.5 Hz, 1H), 8.27–8.20 (m, 3H), 8.17 (d, *J =* 7.9 Hz, 1H), 7.84 (d, *J =* 7.8 Hz, 1H), 7.75 (d, *J =* 8.0 Hz, 1H), 7.71 (d, *J =* 8.0 Hz, 1H), 7.58 (dd, *J =* 8.5, 7.4 Hz, 1H), 7.51 (td, *J =* 7.8, 2.1 Hz, 2H), 3.04 (t, *J =* 7.4 Hz, 2H), 1.80 (*p*, *J =* 7.5 Hz, 2H), 1.45–1.39 (m, 2H), 1.35 (*p*, *J =* 6.5 Hz, 2H), 1.32–1.22 (m, 12H), 0.88 (t, *J =* 7.1 Hz, 3H). ^13^C NMR (CDCl_3_, δ, ppm): δ 204.6, 135.3, 135.1, 134.6, 132.0, 131.3, 131.2, 130.4, 129.4, 129.2, 128.4, 128.3, 128.2, 128.2, 126.9, 126.7, 125.9, 121.7, 121.1, 120.9, 118.9, 42.2, 32.1, 29.8, 29.7, 29.6, 29.6, 29.6, 29.5, 25.1, 22.8, 14.2. HRMS *m*/*z* = 435.2686 [M^+^H]. Calculated *m*/*z* = 435.2682 (C_32_H_35_O^+^).


**General procedure for the synthesis of 3-alkylperylenes (2a–c).**


A mixture of 3-acylperylene **1a–c** (1 equiv), hydrazine monohydrate (50 equiv), and potassium hydroxide (50 equiv) in ethylene glycol (1 g of starting acylperylene in 500 mL) was refluxed with stirring for 24 h. Water was added to the reaction mixture and the organic material was extracted with CH_2_Cl_2_. The organic phase was dried over anhydrous sodium sulfate and evaporated to dryness under vacuum. Pure substances **2a–c** were obtained in the form of a yellow powder by extraction in a Soxhlet extractor with hexane and solvent evaporation.

**3-Butylperylene (2a)** was prepared from 12.0 g (37.2 mmol) of **1a**; yield 5.5 g (48%). Yellow solid. ^1^H NMR (CDCl_3_, δ, ppm, J/Hz): δ 8.42 (d, *J =* 8.4 Hz, 1H), 8.14 (d, *J =* 7.4 Hz, 1H), 8.13–8.08 (m, 2H), 8.04 (d, *J =* 7.9 Hz, 1H), 7.74 (d, *J =* 7.8 Hz, 1H), 7.66 (d, *J =* 8.0 Hz, 1H), 7.62 (d, *J =* 8.0 Hz, 1H), 7.49 (dd, *J =* 8.5, 7.4 Hz, 1H), 7.41 (t, *J =* 7.7 Hz, 2H),3.02–2.97 (m, 2H),1.78–1.71 (m, 2H),1.53–1.46 (m, 2H),1.01 (t, *J =* 7.4 Hz, 3H). ^13^C NMR (CDCl_3_, δ, ppm): δ 135.3, 135.1, 134.6, 132.0, 131.3, 131.1, 130.4, 129.4, 129.2, 128.4, 128.3, 128.2, 128.2, 126.9, 126.7, 125.9, 121.7, 121.1, 120.9, 118.9, 44.1, 31.0, 18.5, 14.1.

**3-Octylperylene (2b)** was prepared from 6.3 g (16.6 mmol) of **1b**; yield 5.6 g (91%). Yellow solid. ^1^H NMR (CDCl_3_, δ, ppm, J/Hz): δ 8.20 (d, *J =* 7.4 Hz, 1H), 8.18–8.08 (m, 3H), 7.88 (d, *J =* 8.3 Hz, 1H), 7.65 (m, 2H), 7.50 (dd, *J =* 8.3, 7.4 Hz, 1H), 7.46 (td, *J =* 7.7, 3.7 Hz, 2H), 7.33 (d, *J =* 7.6 Hz, 1H), 3.04–2.98 (m, 2H), 1.80–1.73 (m, 2H), 1.46 (ddd, *J =* 15.1, 8.4, 6.2 Hz, 2H), 1.41–1.35 (m, 2H), 1.35–1.26 (m, 6H), 0.90 (t, *J =* 7.0 Hz, 3H). ^13^C NMR (CDCl_3_, δ, ppm): δ 139.1, 134.8, 133.2, 131.8, 131.6, 131.6, 129.4, 129.2, 128.6, 127.7, 127.3, 126.8, 126.6, 126.6, 126.3, 123.9, 120.2, 120.1, 120.1, 119.7, 33.5, 32.0, 30.7, 29.9, 29.6, 29.4, 22.7, 14.2. HRMS *m*/*z* = 364.2190 [M^+^]. Calculated *m*/*z* = 364.2186 (C_28_H_28_^+^).

**3-Dodecylperylene (2c)** was prepared from 7.4 g (17.0 mmol) of **1c**; yield 7.0 g (96%). Yellow solid. ^1^H NMR (CDCl_3_, δ, ppm, J/Hz): δ 8.24–8.08 (m, 4H), 7.88 (dd, *J =* 8.4, 1.0 Hz, 1H), 7.68–7.62 (m, 2H), 7.50 (dd, *J =* 8.4, 7.4 Hz, 1H), 7.46 (td, *J =* 7.7, 3.7 Hz, 2H), 7.33 (d, *J =* 7.6 Hz, 1H), 3.03–2.98 (m, 2H), 1.80–1.72 (m, 2H), 1.46 (*p*, *J =* 7.3 Hz, 2H), 1.41–1.33 (m, 2H), 1.33–1.24 (m, 14H), 0.89 (t, *J =* 7.1 Hz, 3H). ^13^C NMR (CDCl_3_, δ, ppm): δ 139.1, 134.8, 133.2, 131.8, 131.6, 131.6, 129.4, 129.2, 128.6, 127.7, 127.3, 126.8, 126.6, 126.6, 126.3, 123.9, 120.1, 120.1, 120.1, 119.7, 61.9, 33.5, 32.0, 30.6, 29.9, 29.8, 29.7, 29.7, 29.7, 29.6, 29.4, 22.7, 14.2. HRMS *m*/*z* = 420.2816 [M^+^]. Calculated *m*/*z* = 420.2812 (C_32_H_36_^+^).


**General procedure for the synthesis of 3-acetyl-9(10)-alkylperylenes (3a–c).**


Acetyl chloride (1.2 equiv) was added to 3-alkylperylene **2a–c** (1 equiv) dissolved in chlorobenzene (10g/L), after which a solution of AlCl_3_ (1.3 equiv) in nitromethane (0.1 g/mL) was added dropwise using an addition funnel. The progress of the reaction was monitored by TLC. After complete conversion of the starting compound, the mixture was poured into a 10% solution of HCl in water, and stirred for 30 min, after which the product was extracted with 3 portions of CH_2_Cl_2_. The organic phase was dried over anhydrous sodium sulfate and evaporated to dryness under vacuum. A pure mixture of isomers of substances **3a–c** in the form of orange crystals was isolated by column chromatography (eluent: chlorobenzene).

At this stage, a mixture of isomers (3-acetyl-9-alkylperylenes and 3-acetyl-10-alkylperylenes) is formed, which can be observed in the ^1^H and ^13^C NMR spectra as the doubling of some signals.

**3-Acelyl-9(10)-butylperylene (3a)** was prepared from 9.9 g (32.1 mmol) of **2a**; yield 7.4 g (66%). Orange solid. ^1^H NMR (CDCl_3_, δ, ppm, J/Hz): δ 8.75–8.67 (m, 1H), 8.28–8.22 (m, 2H), 8.20–8.13 (m, 2H), 8.00–7.91 (m, 2H), 7.63–7.58 (m, 1H), 7.58–7.52 (m, 1H), 7.41–7.35 (m, 1H), 3.08–3.02 (m, 2H), 2.77–2.73 (m, 3H), 1.80–1.72 (m, 2H), 1.52–1.46 (m, 2H), 1.02–0.98 (m, 3H). ^13^C NMR (CDCl_3_, δ, ppm): δ 201.1, 141.1, 136.2, 133.6, 133.0, 132.1, 131.7, 131.6, 130.1, 129.2, 128.7, 128.6, 128.5, 127.0, 126.7, 126.1, 124.3, 122.0, 121.1, 120.9, 118.3, 33.3, 33.0, 30.0, 29.9, 23.0, 14.1. HRMS *m*/*z* = 351.1747 [M + H]^+^. Calculated *m*/*z* = 351.1743 (C_26_H_23_O^+^).

**3-Acelyl-9(10)-octylperylene (3b)** was prepared from 2.6 g (7.2 mmol) of **2b**; yield 2.5 g (65%). Orange solid. ^1^H NMR (CDCl_3_, δ, ppm, J/Hz): δ 8.75–8.66 (m, 1H), 8.24 (dd, *J =* 20.6, 7.5 Hz, 2H), 8.19–8.11 (m, 2H), 8.00–7.90 (m, 2H), 7.63–7.57 (m, 1H), 7.57–7.52 (m, 1H), 7.40–7.35 (m, 1H), 3.07–3.00 (m, 2H), 2.75 (c, 3H), 1.81–1.72 (m, 2H), 1.49–1.44 (m, 2H), 1.42–1.35 (m, 2H), 1.34–1.27 (m, 6H), 0.92–0.86 (m, 3H). ^13^C NMR (CDCl_3_, δ, ppm): δ 201.1, 141.1, 136.2, 133.6, 133.0, 132.1, 131.7, 131.5, 130.1, 129.9, 129.2, 128.6, 128.5, 127.2, 127.0, 126.7, 126.1, 125.6, 125.5, 124.3, 122.0, 121.2, 121.1, 120.9, 120.4, 118.8, 118.3, 33.6, 33.5, 32.1, 30.9, 30.8, 30.1, 30.0, 29.9, 29.5, 22.8, 14.2. HRMS *m*/*z* = 407.2375 [M + H]^+^. Calculated *m*/*z* = 407.2369 (C_30_H_31_O^+^).

**3-Acelyl-9(10)-dodecylperylene (3c)** was prepared from 7.0 g (16.6 mmol) of **2c**; yield 4.1 g (54%). Orange solid. ^1^H NMR (CDCl_3_, δ, ppm, J/Hz): δ 8.73–8.64 (m, 1H), 8.13–8.05 (m, 2H), 8.02–7.76 (m, 4H), 7.54–7.48 (m, 1H), 7.47–7.41 (m, 1H), 7.28–7.23 (m, 1H), 3.00–2.93 (m, 2H), 2.71–2.68 (m, 3H), 1.78–1.70 (m, 2H), 1.49–1.42 (m, 2H), 1.42–1.36 (m, 2H), 1.38–1.27 (m, 14H), 0.91 (t, 3H). ^13^C NMR (CDCl_3_, δ, ppm): δ 200.9, 200.8, 140.8, 139.4, 136.0, 135.9, 133.5, 133.1, 132.7, 131.9, 131.8, 131.4, 131.3, 131.3, 130.6, 130.0, 129.8, 129.0, 128.9, 128.9, 128.5, 128.4, 128.3, 128.3, 128.2, 126.9, 126.7, 126.4, 126.1, 125.9, 125.5, 125.2, 124.0, 121.8, 121.6, 120.9, 120.9, 120.6, 120.1, 118.5, 118.0, 33.5, 33.3, 32.0, 30.7, 30.7, 29.9, 29.9, 29.8, 29.7, 29.7, 29.7, 29.4, 22.8, 14.2. HRMS *m*/*z* = 463.2999 [M + H]^+^. Calculated *m*/*z* = 463.2995 (C_34_H_39_O^+^).


**General procedure for the synthesis of *E/Z* 3-chloro-3-[9(10)-alkylperpylen-3-yl]acroleins (4a–c):**


Chilled POCl_3_ (5 equiv) was added portionwise to DMF and cooled to 0 °C (50 mL per 1 g of the starting compound) with stirring. After 30 min, 3-acetyl-9(10)-alkylperylene **3a–c** (1 equiv) was added to the reaction mixture. The progress of the reaction was monitored by TLC. After one week, the mixture was poured into a 1M solution of NaOAc in water, and stirred for 30 min, after which the product was extracted with 3 portions of CH_2_Cl_2_. The organic phase was dried over anhydrous sodium sulfate and evaporated to dryness under vacuum. A pure mixture of isomers of substances **4a–c** in the form of a red powder was isolated by column chromatography (eluent: toluene), after which the target fractions were evaporated under vacuum.

***E/Z* 3-Chloro-3-[9(10)-butylperpylen-3-yl]acrolein (4a)** was prepared from 5.5 g (15.7 mmol) of **3a**; yield 3.6 g (58%). Red solid; 35:65 E/Z isomer ratio. ^1^H NMR (CDCl_3_, δ, ppm, J/Hz): δ 10.34 (d, *J =* 7.0 Hz, 0.4H), 9.33 (d, *J =* 7.7 Hz, 0.6H), 8.29–8.24 (m, 2H), 8.20–8.12 (m, 2H), 7.98–7.91 (m, 1H), 7.83–7.78 (m, 1H), 7.62–7.53 (m, 2H), 7.53–7.49 (m, 1H), 7.41–7.36 (m, 1H), 6.73 (d, *J =* 7.7 Hz, 0.6H), 6.53 (d, *J =* 7.0 Hz, 0.4H), 3.08–3.03 (m, 2H), 1.80–1.73 (m, 2H), 1.53–1.46 (m, 2H), 1.03–0.98 (m, 3H). ^13^C NMR (CDCl_3_, δ, ppm): δ 191.4, 190.1, 157.3, 151.9, 140.9, 140.8, 134.9, 134.8, 133.8, 133.1, 133.1, 132.9, 132.4, 132.3, 132.1, 131.4, 131.3, 131.2, 130.9, 129.2, 129.1, 128.9, 128.9, 128.5, 128.4, 128.3, 128.1, 127.8, 127.1, 126.7, 124.7, 124.6, 124.6, 124.5, 121.5, 121.2, 121.1, 121.0, 120.9, 118.8, 118.6, 33.3, 33.0, 23.0, 14.1. HRMS *m*/*z* = 397.1359 [M + H]^+^. Calculated *m*/*z* = 397.1353 (C_27_H_22_ClO^+^).

***E/Z* 3-Chloro-3-[9(10)-octylperpylen-3-yl]acrolein (4b)** was prepared from 2.6 g (6.4 mmol) of **3b**; yield 1.7 g (58%). Red solid; 35:65 E/Z isomer ratio ^1^H NMR (CDCl_3_, δ, ppm, J/Hz): δ 10.34 (d, *J =* 7.0 Hz, 0.4H), 9.32 (d, *J =* 7.8 Hz, 0.6H), 8.29–8.06 (m, 4H), 8.01–7.74 (m, 2H), 7.63–7.45 (m, 3H), 7.41–7.31 (m, 1H), 6.73 (d, *J =* 7.8 Hz, 0.6H), 6.53 (d, *J =* 7.0 Hz, 0.4H), 3.09–2.97 (m, 2H), 1.77 (*p*, *J =* 7.4 Hz, 2H), 1.44 (dt, *J =* 12.6, 7.1 Hz, 2H), 1.35–1.23 (m, 8H), 0.94–0.84 (m, 3H). ^13^C NMR (CDCl_3_, δ, ppm): δ 191.1, 189.8, 156.8, 150.9, 140.8, 140.7, 139.9, 134.7, 132.8, 131.1, 129.4, 128.9, 128.0, 127.6, 126.9, 126.5, 124.5, 124.4, 121.4, 121.0, 120.9, 120.3, 119.0, 118.6, 118.4, 33.4, 31.9, 30.7, 29.8, 29.5, 29.3, 22.7, 14.1.

***E/Z* 3-Chloro-3-[9(10)-dodecylperpylen-3-yl]acrolein (4c)** was prepared from 2.3 g (5.0 mmol) of **3c**; yield 1.5 g (59%). Red solid; 35:65 E/Z isomer ratio ^1^H NMR (CDCl_3_, δ, ppm, J/Hz): δ 10.35 (d, *J =* 7.0 Hz, 0.4H), 9.32 (d, 0.6H), 8.25–8.16 (m, 2H), 8.15–8.03 (m, 2H), 7.99–7.76 (m, 2H), 7.62–7.46 (m, 3H), 7.35 (m, 1H), 6.75 (d, *J =* 7.7, 1.4 Hz, 0.4H), 6.53 (d, *J =* 7.0 Hz, 0.6H), 3.03–2.98 (m, 2H), 1.79–1.72 (m, 2H), 1.49–1.42 (m, 2H), 1.39–1.34 (m, 2H), 1.31–1.26 (m, 14H), 0.91–0.87 (m, 3H). ^13^C NMR (CDCl_3_, δ, ppm): δ 191.4, 191.4, 190.1, 157.3, 151.9, 151.9, 140.9, 140.8, 140.0, 134.8, 134.7, 134.6, 134.1, 133.7, 133.0, 133.0, 132.9, 132.3, 132.2, 132.1, 131.3, 131.2, 131.1, 130.8, 129.5, 129.5, 129.0, 128.9, 128.8, 128.8, 128.4, 128.3, 128.1, 127.8, 127.8, 127.7, 127.1, 127.0, 127.0, 126.6, 126.4, 125.3, 124.7, 124.5, 124.5, 124.4, 124.1, 124.0, 121.5, 121.4, 121.1, 121.1, 121.0, 120.9, 120.5, 120.4, 119.1, 119.0, 118.7, 118.5, 33.6, 33.5, 32.1, 30.8, 30.8, 30.0, 29.8, 29.8, 29.8, 29.7, 29.5, 22.8, 14.2. HRMS *m*/*z* = 509.2612 [M + H]^+^. Calculated *m*/*z* = 509.2605 (C_35_H_38_ClO^+^).


**General procedure for the synthesis of 3-ethynyl-9(10)-alkylperylenes (5a–c):**


To a solution of chloroacrolein **4a–c** in a 4:1 *v*/*v* toluene/isopropanol mixture (50 mL per 1 g of compound), KOH (6 equiv) was added. The mixture was evacuated/flushed with argon 5 times and then refluxed with stirring. The progress of the reaction was monitored by TLC. After complete conversion of the starting compound, the mixture was poured into a 10% aqueous HCl solution, stirred for 30 min, and the product was extracted with 3 portions of CH_2_Cl_2_. The organic phase was dried over anhydrous sodium sulfate and evaporated to dryness under vacuum. A pure mixture of isomers of substances **5a–c** was isolated in the form of an orange-brown powder by column chromatography on silica gel, followed by column chromatography on aluminum oxide (eluent: toluene).

**3-Ethynyl-9(10)-butylperylene (5a)** was prepared from 1.7 g (4.3 mmol) of **4a**; yield 1 g (70%). Orange solid. ^1^H NMR (CDCl_3_, δ, ppm, J/Hz): δ 8.19–8.14 (m, 2H), 8.13–8.10 (m, 1H), 8.07–7.98 (m, 2H), 7.89–7.84 (m, 1H), 7.68–7.64 (m, 1H), 7.54–7.49 (m, 1H), 7.49–7.43 (m, 1H), 7.31–7.27 (m, 1H), 3.56–3.52 (m, 1H), 3.02–2.97 (m, 2H), 1.78–1.71 (m, 2H), 1.53–1.46 (m, 2H), 1.03–0.98 (m, 3H). ^13^C NMR (CDCl3, δ, ppm): 140.0, 139.5, 135.0, 133.0, 132.8, 132.7, 132.0, 131.9, 131.9, 131.5, 131.1, 129.1, 128.9, 128.8, 128.8, 128.3, 127.5, 127.4, 126.9, 126.9, 126.4, 126.3, 125.9, 125.5, 124.6, 124.2, 121.0, 121.0, 120.7, 120.6, 120.6, 120.2, 119.4, 118.9, 118.5, 82.8, 82.7, 82.5, 82.5, 33.2, 33.2, 32.9, 32.9, 29.9, 29.8, 23.0, 22.8, 14.3, 14.1.

**3-Ethynyl-9(10)-octylperylene (5b)** was prepared from 0.6 g (1.3 mmol) of **4b**; yield 0.4 g (82%). Orange solid. ^1^H NMR (CDCl_3_, δ, ppm, J/Hz): δ 8.26–8.11 (m, 3H), 8.10–8.00 (m, 2H), 7.90 (dd, *J =* 8.4, 4.0 Hz, 1H), 7.70 (d, *J =* 7.7 Hz, 1H), 7.61–7.47 (m, 2H), 7.36–7.29 (m, 1H), 3.61–3.55 (m, 1H), 3.08–2.97 (m, 2H), 1.87–1.71 (m, 2H), 1.58–1.28 (m, 10H), 0.95 (t, *J =* 6.4 Hz, 3H).^13^C NMR (CDCl_3_, δ, ppm): 140.8, 140.7, 132.9, 132.7, 132.0, 129.4, 128.9, 128.7, 128.0, 127.6, 127.0, 126.9, 126.5, 126.3, 124.5, 124.4, 124.4, 124.3, 124.0, 121.4, 121.3, 121.0, 120.9, 120.9, 120.7, 118.6, 118.4, 82.6, 82.6, 82.5, 82.5, 33.4, 31.9, 30.7, 29.9, 29.5, 29.3, 22.7, 114.1.

**3-Ethynyl-9(10)-dodecylperylene (5c)** was prepared from 0.6 g (1.2 mmol) of **4c**; yield 0.39 g (82%). Orange solid. ^1^H NMR (CDCl_3_, δ, ppm, J/Hz): δ 8.22–8.16 (m, 3H), 8.12–8.03 (m, 2H), 7.92–7.87 (m, 1H), 7.71–7.67 (m, 1H), 7.58–7.48 (m, 2H), 7.35–7.31 (m, 1H), 3.55–3.52 (m, 1H), 3.03–2.98 (m, 2H), 1.79–1.72 (m, 2H), 1.49–1.42 (m, 2H), 1.39–1.34 (m, 2H), 1.31–1.26 (m, 14H), 0.91–0.87 (m, 3H). ^13^C NMR (CDCl_3_, δ, ppm): 140.1, 139.6, 135.0, 133.1, 132.8, 132.8, 132.0, 132.0, 132.0, 131.9, 131.6, 131.2, 129.2, 128.9, 128.8, 128.3, 127.6, 127.5, 127.0, 126.9, 126.5, 126.4, 125.9, 125.5, 124.7, 124.3, 121.1, 121.0, 120.7, 120.7, 120.7, 120.3, 119.4, 119.0, 118.6, 82.8, 82.7, 82.5, 82.4, 33.6, 33.5, 32.1, 30.8, 30.8, 30.0, 29.8, 29.8, 29.7, 29.5, 22.8, 14.3.


**General procedure for the synthesis of *O*-propionyl protected 5-[9(10)-alkylperylen-3-ylethynyl)uracil nucleosides (6–8a–c):**


The corresponding *O*-propionylated 5-iodo-(*deoxy*/*ribo*/*arabino*)uridine (1 equiv) and 3-ethynyl-9(10)-alkylperylene **5a–c** (1.3 equiv) were dissolved in dry DMF (3 mL/25 mg of starting ethynyl compound) in a Schlenk flask and the mixture was evacuated and flushed with argon five times to remove traces of oxygen. Then tetrakis(triphenylphosphine)palladium(0) (0.1 equiv) and triethylamine (2 equiv) were added, the mixture was evacuated and flushed with argon 3 times, after which copper iodide (0.4 equiv) was added, and the mixture was evacuated and flushed with argon 3 times. The Schlenk flask was then heated at 80 °C in a glycerol bath and the mixture was stirred in the dark for 3 h. After the almost complete disappearance of the starting product, the mixture was poured into 2.5% (*w/w*, aq.) disodium EDTA solution. The mixture was extracted with 3 portions of ethyl acetate. The organic layer was thoroughly washed with disodium EDTA, water (4 times), and brine and dried over anhydrous Na_2_SO_4_. The solvent was removed in vacuo. The residue was purified by column chromatography on silica gel with a gradient of EtOAc in CH_2_Cl_2_ (0→8%, *v*/*v*) to afford the title product as an orange solid.

**2′,3′,5′-*Tris*-*O*-propionyl-5-[9(10)-butylperylen-3-ylethynyl]-*arabino*-uridine (6a)** was prepared from 25 mg (75.2 µmol) of **5a**; yield 26 mg (60%). Orange solid. ^1^H NMR (CDCl_3_, δ, ppm, J/Hz): δ 8.35–8.30 (m, 1H), 8.27–8.21 (m, 3H), 8.19–8.09 (m, 2H), 7.95–7.90 (m, 2H), 7.73–7.69 (m, 1H), 7.62–7.58 (m, 1H), 7.56–7.51 (m, 1H), 7.38–7.35 (m, 1H), 6.35 (d, *J =* 4.3 Hz, 1H), 5.51 (dd, *J =* 4.3, 2.0 Hz, 1H), 5.19 (dd, *J =* 3.9, 2.0 Hz, 1H), 4.56–4.43 (m, 2H), 4.27–4.23 (m, 1H), 3.06–3.01 (m, 2H), 2.49–2.34 (m, 6H), 1.78–1.72 (m, 2H), 1.52–1.45 (m, 2H), 1.21–1.12 (m, 9H), 0.99 (t, *J =* 7.4 Hz, 3H). ^13^C NMR (CDCl_3_, δ, ppm): δ 174.1, 173.3, 172.4, 160.5, 148.7, 142.1, 140.2, 134.7, 133.1, 132.9, 131.9, 131.6, 131.3, 131.2, 128.9, 128.9, 128.4, 127.8, 127.7, 127.1, 127.0, 126.6, 126.4, 126.2, 124.3, 121.2, 121.1, 120.9, 120.8, 120.7, 120.4, 119.5, 119.1, 118.9, 100.5, 93.3, 85.5, 84.7, 80.9, 76.1, 74.7, 62.5, 33.3, 33.2, 33.0, 33.0, 29.9, 27.6, 27.5, 27.4, 23.0, 14.2, 9.2, 9.0. HRMS *m*/*z* = 743.2967 [M + H]^+^. Calculated *m*/*z* = 743.2963 (C_44_H_43_N_2_O_9_^+^).

**2′,3′,5′-*Tris*-*O*-propionyl-5-[9(10)-butylperylen-3-ylethynyl]-*ribo*-uridine (7a)** was prepared from 25 mg (75.2 µmol) of **5a**; yield 18 mg (46%). Orange solid. ^1^H NMR (CDCl_3_, δ, ppm, J/Hz): 8.33–8.21 (m, 3H), 8.17–8.09 (m, 2H), 7.95–7.90 (m, 3H), 7.68–7.64 (m, 1H), 7.62–7.57 (m, 1H), 7.56–7.52 (m, 1H), 7.38–7.35 (m, 1H), 6.18 (d, *J =* 5.4 Hz, 1H), 5.45–5.41 (m, 1H), 5.42–5.39 (m, 1H), 4.45–4.39 (m, 3H), 3.04 (t, 2H), 2.54–2.49 (m, 2H), 2.46–2.33 (m, 4H), 1.78–1.72 (m, 2H), 1.51–1.46 (m, 2H), 1.20–1.12 (m, 6H), 1.09 (t, *J =* 7.4 Hz, 3H), 0.99 (t, *J =* 7.4 Hz, 3H). ^13^C NMR (CDCl_3_, δ, ppm): 173.7, 173.2, 173.2, 160.5, 149.1, 140.7, 133.1, 133.0, 131.9, 131.6, 131.3, 128.9, 127.8, 127.0, 126.6, 124.3, 121.2, 120.9, 120.8, 119.1, 102.0, 87.4, 80.7, 73.3, 70.3, 63.1, 51.0, 33.3, 33.2, 33.0, 33.0, 29.9, 27.7, 27.4, 27.3, 23.0, 14.2, 9.1, 9.0. HRMS *m*/*z* = 743.2979 [M + H]^+^. Calculated *m*/*z* = 743.2963 (C_44_H_42_N_2_O_9_^+^).

**3′,5′-*Bis*-*O*-propionyl-5-[9(10)-butylperylen-3-ylethynyl]-*deoxy*-uridine (8a)** was prepared from 25 mg (75.2 µmol) of **5a**, yield 18 mg (47%). Orange solid. ^1^H NMR (CDCl_3_, δ, ppm, J/Hz): 8.32–8.25 (m, 1H), 8.26–8.21 (m, 1H), 8.23–8.18 (m, 1H), 8.18–8.10 (m, 1H), 8.11–8.06 (m, 2H), 7.98–7.96 (m, 1H), 7.94–7.88 (m, 1H), 7.67–7.63 (m, 1H), 7.62–7.55 (m, 1H), 7.57–7.49 (m, 1H), 7.38–7.32 (m, 1H), 6.40–6.33 (m, 1H), 5.31–5.27 (m, 1H), 4.52–4.39 (m, 2H), 4.34–4.30 (m, 1H), 3.06–3.00 (m, 2H), 2.62–2.56 (m, 1H), 2.53–2.32 (m, 4H), 2.31–2.23 (m, 1H), 1.78–1.71 (m, 2H), 1.53–1.44 (m, 2H), 1.18 (t, *J =* 7.6 Hz, 3H), 1.09 (t, *J =* 7.5 Hz, 3H), 0.99 (t, *J =* 7.5 Hz, 3H). ^13^C NMR (CDCl_3_, δ, ppm): 174.0, 173.8, 160.2, 149.2, 140.8, 140.2, 134.7, 133.1, 132.9, 131.9, 131.6, 131.1, 131.1, 128.9, 128.9, 128.4, 127.8, 127.8, 127.7, 127.1, 127.0, 126.5, 126.4, 126.1, 124.3, 121.4, 121.1, 120.9, 120.7, 119.5, 119.2, 119.1, 118.8, 102.1, 101.5, 101.5, 93.4, 88.8, 85.8, 85.7, 83.7, 83.0, 74.2, 74.1, 63.9, 63.8, 51.0, 39.6, 38.6, 33.3, 33.2, 32.9, 32.9, 29.8, 27.7, 27.6, 23.0, 22.8, 14.2, 14.1, 9.2, 9.1. HRMS *m*/*z* = 671.2761 [M + H]^+^. Calculated *m*/*z* = 671.2752 (C_41_H_39_N_2_O_7_^+^).

**2′,3′,5′-*Tris*-*O*-propionyl-5-[9(10)-octylperylen-3-ylethynyl]-*arabino*-uridine (6b)** was prepared from 50 mg (128.7 µmol) of **5b**; yield 58 mg (73%). Orange solid. ^1^H NMR (CDCl_3_, δ, ppm, J/Hz): δ 8.36–8.29 (m, 1H), 8.26–8.15 (m, 3H), 8.14–8.04 (m, 2H), 7.93–7.82 (m, 2H), 7.71–7.66 (m, 1H), 7.62–7.47 (m, 2H), 7.35–7.30 (m, 1H), 6.37–6.33 (m, 1H), 5.53–5.49 (m, 1H), 5.22–5.18 (m, 1H), 4.56–4.43 (m, 2H), 4.26–4.21 (m, 1H), 3.04–2.97 (m, 2H), 2.50–2.32 (m, 6H), 1.81–1.71 (m, 2H), 1.48–1.41 (m, 2H), 1.41–1.34 (m, 2H), 1.34–1.26 (m, 6H), 1.25–1.11 (m, 9H), 0.89 (t, *J =* 6.7 Hz, 3H). ^13^C NMR (CDCl_3_, δ, ppm): 174.4, 174.1, 173.4, 173.3, 160.7, 148.8, 142.0, 140.1, 139.6, 134.7, 133.1, 133.0, 132.8, 132.8, 131.9, 131.6, 131.3, 131.2, 131.1, 129.2, 128.9, 128.9, 128.4, 127.7, 127.7, 127.0, 126.9, 126.5, 126.4, 126.2, 125.8, 124.2, 121.1, 121.0, 120.9, 120.7, 120.7, 120.4, 119.5, 119.4, 119.1, 118.9, 100.5, 93.3, 93.3, 85.7, 85.6, 84.7, 82.0, 80.9, 76.2, 74.7, 62.9, 62.5, 33.6, 33.5, 32.0, 30.8, 30.0, 29.7, 29.4, 27.7, 27.6, 27.5, 27.4, 22.8, 14.2, 9.2, 9.0. HRMS *m*/*z* = 799.3599 [M + H]^+^. Calculated *m*/*z* = 799.3589 (C_48_H_51_N_2_O_9_^+^).

**2′,3′,5′-*Tris*-*O*-propionyl-5-[9(10)-octylperylen-3-ylethynyl]-*ribo*-uridine (7b)** was prepared from 200 mg (514.7 µmol) of **5b**; yield 168 mg (58%). Orange solid. ^1^H NMR (CDCl_3_, δ, ppm, J/Hz): 8.30–8.23 (m, 1H), 8.22–8.13 (m, 2H), 8.11–8.00 (m, 3H), 7.91 (d, *J =* 1.7 Hz, 1H), 7.87 (t, *J =* 8.6 Hz, 1H), 7.63–7.55 (m, 2H), 7.52–7.45 (m, 1H), 7.34–7.28 (m, 1H), 6.19 (d, *J =* 5.4 Hz, 1H), 5.45 (t, *J =* 5.6 Hz, 1H), 5.42 (dd, *J =* 5.7, 3.7 Hz, 1H), 4.46–4.39 (m, 3H), 3.02–2.96 (m, 2H), 2.55–2.48 (m, 2H), 2.47–2.34 (m, 4H), 1.78–1.70 (m, 2H), 1.48–1.41 (m, 2H), 1.49–1.40 (m, 0H), 1.39–1.34 (m, 2H), 1.34–1.24 (m, 6H), 1.17 (dt, *J =* 14.8, 7.6 Hz, 6H), 1.10 (t, *J =* 7.5 Hz, 3H), 0.89 (t, *J =* 6.9 Hz, 3H).^13^C NMR (CDCl_3_, δ, ppm): 173.8, 173.2, 173.2, 160.7, 149.3, 140.6, 140.1, 134.7, 133.1, 132.9, 131.8, 131.6, 131.1, 128.9, 128.8, 128.3, 127.8, 127.7, 127.0, 126.9, 126.5, 126.3, 126.2, 124.2, 121.1, 120.8, 120.7, 120.6, 120.4, 119.0, 118.7, 102.1, 93.7, 87.4, 85.5, 80.7, 73.3, 70.3, 63.1, 33.6, 33.5, 32.0, 30.8, 30.8, 30.0, 29.7, 29.5, 27.7, 27.4, 27.3, 22.8, 14.2, 9.1, 9.0. HRMS *m*/*z* = 799.3605 [M + H]^+^. Calculated *m*/*z* = 799.3589 (C_48_H_51_N_2_O_9_^+^).

**3′,5′-*Bis*-*O*-propionyl-5-[9(10)-octylperylen-3-ylethynyl]-*deoxy*-uridine (8b)** was prepared from 200 mg (514.7 µmol) of **5b**; yield 140 mg (60%). Orange solid. ^1^H NMR (CDCl_3_, δ, ppm, J/Hz): δ 8.28–8.11 (m, 4H), 8.10–7.98 (m, 2H), 7.90–7.83 (m, 1H), 7.80–7.74 (m, 1H), 7.61–7.43 (m, 3H), 7.37–7.26 (m, 1H), 6.80 (d, *J =* 5.4 Hz, 1H), 6.41–6.31 (m, 1H), 5.29 (d, *J =* 6.4 Hz, 1H), 4.51–4.38 (m, 3H), 4.34–4.28 (m, 1H), 3.04–2.95 (m, 2H), 2.52–2.30 (m, 4H), 1.79–1.70 (m, 2H), 1.50–1.41 (m, 2H), 1.40–1.34 (m, 2H), 1.34–1.24 (m, 6H), 1.23–1.06 (m, 6H), 0.95–0.86 (m, 3H). ^13^C NMR (CDCl_3_, δ, ppm): 174.0, 174.0, 173.8, 171.9, 155.9, 154.7, 140.7, 140.4, 140.1, 139.7, 139.5, 135.0, 135.0, 134.6, 133.8, 133.7, 133.0, 133.0, 132.8, 132.7, 132.2, 132.2, 131.9, 131.6, 131.5, 131.0, 131.0, 129.1, 129.0, 128.7, 128.6, 128.3, 128.0, 127.7, 126.9, 126.5, 126.3, 124.9, 124.8, 124.6, 124.4, 124.3, 124.2, 124.0, 121.3, 121.2, 121.1, 120.9, 120.9, 120.7, 120.6, 120.2, 119.6, 119.4, 119.1, 119.0, 108.4, 102.2, 102.0, 88.8, 85.9, 85.8, 83.7, 83.1, 76.9, 74.2, 74.1, 63.8, 63.8, 53.5, 39.6, 38.5, 33.5, 33.5, 32.0, 30.8, 30.8, 30.0, 30.0, 29.7, 29.4, 27.7, 27.7, 27.6, 27.6, 22.8, 14.2, 9.2, 9.1.

**2′,3′,5′-*Tris*-*O*-propionyl-5-[9(10)-dodecylperylen-3-ylethynyl]-*arabino*-uridine (6c)** was prepared from 20 mg (45.0 µmol) of **5c**; yield 17 mg (57%). Orange solid. ^1^H NMR (CDCl_3_, δ, ppm, J/Hz): δ 8.84 (c, 1H), 8.33 (dd, *J =* 14.9, 8.2 Hz, 1H), 8.25–8.17 (m, 2H), 8.15–8.06 (m, 2H), 7.92 (c, 1H), 7.92–7.87 (m, 1H), 7.72–7.67 (m, 1H), 7.62–7.57 (m, 1H), 7.55–7.48 (m, 1H), 7.35–7.31 (m, 1H), 6.35 (d, *J =* 4.3 Hz, 1H), 5.53–5.50 (m, 1H), 5.21–5.18 (m, 1H), 4.55–4.50 (m, 1H), 4.49–4.44 (m, 1H), 4.26–4.21 (m, 1H), 3.00 (t, *J =* 7.9 Hz, 2H), 2.50–2.35 (m, 6H), 1.79–1.72 (m, 2H), 1.47–1.42 (m, 2H), 1.40–1.33 (m, 2H), 1.32–1.25 (m, 14H), 1.24–1.13 (m, 9H), 0.88 (t, *J =* 7.0 Hz, 3H). ^13^C NMR (CDCl_3_, δ, ppm): δ 174.3, 173.3, 172.0, 171.7, 156.0, 154.3, 140.5, 136.8, 133.9, 133.0, 132.3, 131.6, 131.5, 129.0, 128.7, 128.6, 128.1, 127.8, 127.0, 126.5, 126.5, 124.4, 124.4, 121.4, 121.3, 121.0, 120.8, 120.3, 119.6, 119.2, 108.1, 102.0, 87.3, 82.0, 76.6, 74.1, 62.9, 53.5, 33.6, 33.5, 32.1, 30.8, 30.0, 29.8, 29.8, 29.7, 29.5, 27.6, 27.4, 27.4, 22.8, 14.2, 9.2, 9.0, 8.9. HRMS *m*/*z* = 855.4221 [M + H]^+^. Calculated *m*/*z* = 855.4215 (C_52_H_59_N_2_O_9_^+^).

**2′,3′,5′-*Tris*-*O*-propionyl-5-[9(10)-dodecylperylen-3-ylethynyl]-*ribo*-uridine (7c)** was prepared from 20 mg (45.0 µmol) of **5c**; yield 25 mg (84%). Orange solid. ^1^H NMR (CDCl_3_, δ, ppm, J/Hz): δ 8.65–8.63 (m, 1H), 8.30–8.24 (m, 1H), 8.23–8.15 (m, 2H), 8.13–8.02 (m, 2H), 7.93–7.90 (m, 1H), 7.91–7.86 (m, 1H), 7.64–7.60 (m, 1H), 7.61–7.55 (m, 1H), 7.54–7.47 (m, 1H), 7.35–7.30 (m, 1H), 6.19 (d, *J =* 5.4 Hz, 1H), 5.45 (t, *J =* 5.6 Hz, 1H), 5.43–5.39 (m, 1H), 4.46–4.39 (m, 3H), 3.02–2.97 (m, 2H), 2.55–2.49 (m, 2H), 2.47–2.34 (m, 4H), 1.78–1.71 (m, 2H), 1.48–1.39 (m, 2H), 1.40–1.33 (m, 2H), 1.33–1.23 (m, 14H), 1.18 (dt, *J =* 14.9, 7.6 Hz, 6H), 1.10 (t, *J =* 7.5 Hz, 3H), 0.88 (t, *J =* 7.0 Hz, 3H). ^13^C NMR (CDCl_3_, δ, ppm): δ 173.9, 173.7, 173.2, 173.2, 172.2, 160.6, 160.6, 156.4, 154.8, 140.7, 140.7, 140.5, 140.2, 139.6, 135.0, 134.7, 134.0, 133.1, 133.0, 132.9, 132.8, 132.6, 132.2, 131.9, 131.7, 131.5, 131.2, 131.2, 131.1, 131.0, 129.2, 129.0, 129.0, 128.9, 128.9, 128.8, 128.8, 128.6, 128.3, 128.2, 127.8, 127.7, 127.0, 126.9, 126.5, 126.5, 126.4, 126.1, 125.7, 124.7, 124.5, 124.4, 124.4, 124.2, 121.4, 121.1, 121.1, 120.7, 120.7, 120.4, 119.4, 119.2, 119.1, 119.0, 118.7, 109.2, 102.0, 102.0, 101.9, 93.6, 93.6, 89.9, 87.4, 85.6, 85.5, 80.7, 80.3, 74.3, 73.3, 70.3, 69.7, 68.3, 63.1, 62.9, 53.6, 38.9, 33.6, 33.6, 33.5, 32.1, 30.8, 30.8, 30.5, 30.0, 30.0, 29.8, 29.8, 29.7, 29.5, 29.1, 27.7, 27.7, 27.4, 27.4, 27.4, 27.3, 23.9, 23.1, 22.8, 14.3, 14.2, 11.1, 9.2, 9.1, 9.0, 9.0. HRMS *m*/*z* = 855.4222 [M + H]^+^. Calculated *m*/*z* = 855.4215 (C_52_H_59_N_2_O_9_^+^).

**3′,5′-*Bis*-*O*-propionyl-5-[9(10)-dodecylperylen-3-ylethynyl]-*deoxy*-uridine (8c)** was prepared from 20 mg (45.0 µmol) of **5c**; yield 22 mg (81%). Orange solid. ^1^H NMR (CDCl_3_, δ, ppm, J/Hz): δ 8.82 (c, 1H), 8.34–8.26 (m, 1H), 8.23–8.16 (m, 2H), 8.13–8.03 (m, 2H), 7.96 (c, 1H), 7.92–7.86 (m, 1H), 7.65–7.61 (m, 1H), 7.61–7.56 (m, 1H), 7.55–7.47 (m, 1H), 7.35–7.30 (m, 1H), 6.39–6.33 (m, 1H), 5.31–5.27 (m, 1H), 4.49–4.43 (m, 1H), 4.44–4.39 (m, 1H), 4.33–4.30 (m, 1H), 3.05–2.97 (m, 2H), 2.62–2.56 (m, 2H), 2.52–2.35 (m, 4H), 1.78–1.71 (m, 2H), 1.48–1.41 (m, 2H), 1.39–1.34 (m, 2H), 1.36–1.25 (m, 14H), 1.18 (t, *J =* 7.6 Hz, 3H), 1.09 (t, *J =* 7.5 Hz, 3H), 0.88 (t, *J =* 7.1 Hz, 3H). ^13^C NMR (CDCl_3_, δ, ppm): δ 174.1, 174.0, 171.9, 155.9, 154.6, 140.4, 134.9, 133.8, 133.0, 131.6, 131.5, 129.0, 128.7, 128.6, 128.1, 127.7, 127.0, 126.5, 124.4, 124.4, 121.3, 120.9, 120.7, 119.2, 108.4, 102.0, 88.8, 83.7, 74.3, 63.8, 39.6, 33.6, 32.1, 30.8, 30.0, 29.8, 29.8, 29.7, 29.5, 27.7, 27.6, 22.8, 14.2, 9.2, 9.0. HRMS *m*/*z* = 781.3855 [M − H]^−^. Calculated *m*/*z* = 781.3848 (C_49_H_53_N_2_O_7_^−^).


**General procedure for the synthesis of 5-[9(10)-alkylperylen-3-ylethynyl)uracil nucleosides (9–11a–c):**


K_2_CO_3_ (15 equiv) was added to compound **6–8a–c** (1 equiv) dissolved in a CH_2_Cl_2_/MeOH/H_2_O mixture (1:5:1 *v*/*v*/*v*, 3 mL per 10 mg of a starting compound) and stirred at room temperature for 48 h. After that, the mixture was transferred to polypropylene microcentrifuge tubes, centrifuged, suspended in 10% hydrochloric acid, and centrifuged again. The resulting precipitate was suspended in water and centrifuged (5 times), then suspended in ethyl acetate and centrifuged (2 times). The product was obtained as an orange-brown powder by drying the precipitate under vacuum.

**5-[9(10)-Butylperylen-3-ylethynyl]-*arabino*-uridine (9a).** Yield 9 mg (95%). Orange solid. ^1^H NMR (DMSO-*d*_6_, δ, ppm, J/Hz): δ 8.47–8.21 (m, 7H), 8.00–7.95 (m, 1H), 7.73–7.64 (m, 2H), 7.63–7.58 (m, 1H), 7.42 (d, *J =* 7.7 Hz, 1H), 6.06 (d, *J =* 4.6 Hz, 1H), 5.68 (c, 1H), 5.50 (c, 1H), 5.22 (c, 1H), 4.10–4.08 (m, 1H), 4.01–3.97 (m, 1H), 3.83–3.78 (m, 1H), 3.73–3.65 (m, 2H), 3.04–2.99 (m, 2H), 1.71–1.64 (m, 2H), 1.47–1.39 (m, 2H), 0.95 (t, *J =* 7.4 Hz, 3H). ^13^C NMR (DMSO-*d*_6_, δ, ppm): δ 170.0, 155.1, 149.9, 145.5, 145.4, 140.1, 139.8, 134.2, 132.9, 131.9, 131.7, 131.0, 130.9, 128.8, 128.5, 128.5, 128.4, 128.4, 128.4, 128.0, 127.6, 127.6, 127.3, 127.3, 126.1, 125.7, 124.8, 121.9, 121.8, 121.6, 121.6, 120.2, 119.9, 119.5, 97.5, 97.5, 91.1, 91.0, 89.4, 89.2, 86.1, 85.3, 75.8, 75.5, 60.8, 32.9, 32.9, 32.7, 32.6, 22.7, 14.3. HRMS *m*/*z* = 573.2024 [M − H]^−^. Calculated *m*/*z* = 573.2020 (C_35_H_29_N_2_O_6_^+^).

**5-[9(10)-Butylperylen-3-ylethynyl]-*ribo*-uridine (10a).** Yield 8 mg (95%). Orange solid. ^1^H NMR (DMSO-*d*_6_, δ, ppm, J/Hz): δ 8.44–8.25 (m, 7H), 7.96 (d, *J =* 8.4 Hz, 1H), 7.68–7.61 (m, 2H), 77.62–7.57 (m, 1H), 7.41 (d, *J =* 7.7 Hz, 1H), 5.83 (d, *J =* 4.5 Hz, 1H), 5.42 (c, 1H), 5.29 (c, 1H), 5.12 (c, 1H), 4.12–4.08 (m, 1H), 4.08–4.03 (m, 1H), 3.89–3.85 (m, 1H), 3.68 (dd, *J =* 75.7, 11.5 Hz, 2H), 3.01 (t, *J =* 7.9 Hz, 2H), 1.71–1.64 (m, 2H), 1.48–1.40 (m, 2H), 0.96 (t, *J =* 7.4 Hz, 3H). ^13^C NMR (DMSO-*d*_6_, δ, ppm): δ 143.0, 139.2, 139.0, 133.7, 132.4, 131.0, 130.8, 130.5, 129.8, 128.4, 128.2, 128.0, 127.6, 127.5, 127.1, 126.8, 126.2, 124.3, 124.1, 121.2, 121.1, 120.9, 120.9, 120.7, 120.2, 119.8, 98.3, 98.2, 89.1, 84.3, 79.1, 73.9, 69.4, 60.5, 32.4, 32.2, 32.1, 22.2, 13.8. HRMS *m*/*z* = 574.2122 [M^+^]. Calculated *m*/*z* = 574.2098 (C_35_H_30_N_2_O_6_^+^).

**5-[9(10)-Butylperylen-3-ylethynyl]-*deoxy*-uridine (11a).** Yield 8 mg (95%). Orange solid. ^1^H NMR (DMSO-*d*_6_, δ, ppm, J/Hz): δ 8.52 (d, *J =* 3.4 Hz, 1H), 8.50–8.40 (m, 2H), 8.40–8.19 (m, 3H), 8.03–7.96 (m, 1H), 7.72–7.59 (m, 3H), 7.48–7.42 (m, 1H), 7.27 (d, *J =* 7.2 Hz, 1H), 6.20–6.15 (m, 1H), 5.27 (d, *J =* 4.3 Hz, 1H), 5.22 (t, *J =* 5.0 Hz, 1H), 4.33–4.27 (m, 1H), 3.99–3.95 (m, 1H), 3.92–3.83 (m, 2H), 3.75–3.61 (m, 2H), 3.03 (q, *J =* 7.3 Hz, 2H), 1.71–1.64 (m, 2H), 1.44 (h, *J =* 7.3 Hz, 2H), 0.96 (t, *J =* 7.4, 3H). ^13^C NMR (DMSO-*d*_6_, δ, ppm): δ 171.0, 162.9, 161.5, 149.4, 143.6, 139.7, 138.4, 133.7, 132.4, 131.4, 131.2, 130.6, 130.5, 128.0, 127.9, 127.5, 127.1, 126.8, 125.7, 124.6, 121.5, 121.3, 121.2, 119.7, 118.9, 106.7, 103.9, 98.4, 87.7, 87.6, 85.0, 69.8, 69.7, 60.8, 60.8, 41.3, 40.2, 40.0, 32.4, 32.4, 32.2, 32.1, 22.2, 22.2, 13.8. HRMS *m*/*z* = 558.2159 [M^+^]. Calculated *m*/*z* = 558.2149 (C_35_H_30_N_2_O_6_^+^).

**5-[9(10)-Octylperylen-3-ylethynyl]-*arabino*-uridine (9b).** Yield 50 mg (95%). Orange solid. ^1^H NMR (DMSO-*d*_6_, δ, ppm, J/Hz): δ 8.42–8.27 (m, 4H), 8.27–8.20 (m, 2H), 8.09–8.06 (m, 1H), 7.92 (d, *J =* 8.3 Hz, 1H), 7.68–7.61 (m, 2H), 7.59–7.53 (m, 1H), 7.38–7.34 (m, 1H), 6.12–6.09 (m, 1H), 5.65 (c, 2H), 5.31 (c, 1H), 4.06–3.98 (m, 2H), 3.79–3.75 (m, 1H), 3.71–3.68 (m, 2H), 2.99–2.94 (m, 2H), 1.69–1.61 (m, 2H), 1.42–1.34 (m, 2H), 1.34–1.28 (m, 2H), 1.27–1.20 (m, 6H), 0.84 (t, *J =* 6.7 Hz, 3H). ^13^C NMR (DMSO-*d*_6_, δ, ppm): δ 171.0, 154.2, 152.8, 152.1, 144.5, 139.2, 139.0, 133.7, 132.4, 131.0, 130.7, 130.6, 130.5, 129.8, 128.3, 128.1, 127.9, 127.5, 127.5, 127.0, 126.7, 126.0, 125.6, 124.0, 121.1, 121.0, 120.8, 120.8, 120.6, 120.1, 119.9, 119.7, 96.9, 89.5, 85.3, 84.5, 75.6, 75.4, 60.7, 32.4, 31.2, 30.1, 29.0, 28.8, 28.6, 22.0, 13.8. HRMS *m*/*z* = 653.2627 [M + Na]^+^. Calculated *m*/*z* = 653.2622 (C_39_H_38_N_2_O_6_Na^+^).

**5-[9(10)-Octylperylen-3-ylethynyl]-*ribo*-uridine (10b).** Yield 34 mg (95%). Orange solid. ^1^H NMR (DMSO-*d*_6_, δ, ppm, J/Hz): δ 8.52 (c, 1H), 8.40 (ddd, *J =* 22.0, 14.6, 7.2 Hz, 3H), 8.35–8.25 (m, 3H), 7.97–7.93 (m, 1H), 7.70–7.64 (m, 2H), 7.61–7.57 (m, 1H), 7.45–7.36 (m, 1H), 5.83 (d, *J =* 4.4 Hz, 1H), 5.61–5.41 (m, 1H), 5.38 (c, 1H), 5.20 (c, 1H), 4.16–4.12 (m, 1H), 4.11–4.06 (m, 1H), 3.93–3.89 (m, 1H), 3.79–3.74 (m, 1H), 3.68–3.63 (m, 1H), 3.02–2.97 (m, 2H), 1.70–1.64 (m, 2H), 1.39 (*p*, *J =* 7.4 Hz, 2H), 1.35–1.29 (m, 2H), 1.28–1.22 (m, 6H), 0.87–0.82 (m, 3H). ^13^C NMR (DMSO-*d*_6_, δ, ppm): δ 161.2, 153.7, 143.6, 139.5, 133.7, 132.4, 131.1, 131.1, 130.7, 130.4, 130.2, 130.2, 129.0, 128.4, 128.3, 127.9, 127.5, 127.1, 126.8, 121.3, 121.0, 119.7, 105.7, 104.5, 102.2, 98.4, 90.2, 88.9, 84.5, 74.0, 69.2, 60.3, 32.5, 31.2, 30.2, 29.1, 28.8, 28.7, 22.0, 13.9. HRMS *m*/*z* = 629.2652 [M − H]^−^. Calculated *m*/*z* = 629.2646 (C_39_H_37_N_2_O_6_^−^).

**5-[9(10)-Octylperylen-3-ylethynyl]-*deoxy*-uridine (11b).** Yield 45 mg (95%). Orange solid. ^1^H NMR (DMSO-*d*_6_, δ, ppm, J/Hz): δ 8.49 (c, 1H), 8.44–8.36 (m, 2H), 8.36–8.24 (m, 2H), 7.97–7.92 (m, 1H), 7.67 (dq, *J =* 9.8, 6.6, 5.3 Hz, 1H), 7.61–7.56 (m, 1H), 7.41–7.37 (m, 1H), 6.19 (t, *J =* 6.5 Hz, 1H), 5.25 (c, 2H), 4.34–4.29 (m, 1H), 4.26–4.15 (m, 1H), 3.88–3.83 (m, 2H), 3.74–3.62 (m, 2H), 2.98 (t, *J =* 7.6 Hz, 2H), 2.28–2.17 (m, 2H), 1.70–1.62 (m, 2H), 1.43–1.34 (m, 3H), 1.34–1.28 (m, 3H), 1.29–1.16 (m, 6H), 0.83 (t, *J =* 6.6 Hz, 3H). ^13^C NMR (DMSO-*d*_6_, δ, ppm): δ 172.1, 162.0, 161.0, 149.8, 143.5, 143.4, 139.5, 139.2, 133.7, 132.4, 131.3, 131.2, 131.1, 131.1, 130.6, 130.3, 130.3, 128.2, 128.0, 127.9, 127.8, 127.8, 127.5, 127.0, 127.0, 126.7, 126.7, 125.7, 125.3, 124.5, 124.2, 121.3, 121.2, 121.0, 121.0, 120.7, 120.1, 119.6, 119.4, 119.0, 98.4, 98.3, 90.5, 90.5, 89.0, 87.6, 84.9, 69.8, 60.8, 32.4, 32.3, 31.2, 30.1, 29.0, 28.8, 28.6, 22.0, 13.8. HRMS *m*/*z* = 614.2786 [M^+^]. Calculated *m*/*z* = 614.2775 (C_39_H_37_N_2_O_5_^+^).

**5-[9(10)-Dodecylperylen-3-ylethynyl]-*arabino*-uridine (9c).** Yield 11 mg (95%). Orange solid. ^1^H NMR (DMSO-*d*_6_, δ, ppm, J/Hz): δ 8.65–8.63 (m, 1H), 8.50–8.41 (m, 3H), 8.38–8.34 (m, 1H), 8.26–8.20 (m, 1H), 8.02–7.97 (m, 1H), 7.92–7.88 (m, 1H), 7.70–7.65 (m, 1H), 7.65–7.60 (m, 1H), 7.46–7.43 (m, 1H), 7.32–7.28 (m, 1H), 6.24 (d, *J =* 3.9 Hz, 1H), 5.56–5.53 (m, 2H), 5.11 (t, *J =* 5.6 Hz, 1H), 4.20 (ddd, *J =* 5.6, 3.9, 2.0 Hz, 1H), 4.01 (dt, *J =* 4.4, 2.3 Hz, 1H), 3.95 (td, *J =* 5.7, 2.7 Hz, 1H), 3.74–3.67 (m, 2H), 3.04–3.00 (m, 2H), 1.72–1.66 (m, 2H), 1.43–1.37 (m, 2H), 1.35–1.29 (m, 2H), 1.27–1.18 (m, 14H), 0.83 (t, *J =* 7.1 Hz, 3H).^13^C NMR (DMSO-*d*_6_, δ, ppm): δ 171.0, 153.8, 152.7, 152.7, 140.3, 140.3, 139.9, 139.3, 132.4, 132.3, 131.3, 130.8, 130.7, 130.3, 128.4, 128.1, 128.0, 127.9, 127.9, 127.8, 127.2, 126.9, 126.8, 125.1, 124.7, 124.6, 124.3, 121.6, 121.3, 121.2, 120.7, 120.2, 119.8, 106.1, 104.2, 104.0, 88.4, 86.3, 76.2, 74.5, 61.1, 32.5, 32.4, 31.2, 30.2, 29.0, 29.0, 28.9, 28.8, 28.6, 22.0, 13.9. HRMS *m*/*z* = 709.3248 [M + Na]^+^. Calculated *m*/*z* = 709.3248 (C_43_H_46_N_2_O_6_Na^+^).

**5-[9(10)-Dodecylperylen-3-ylethynyl]-*ribo*-uridine (10c).** Yield 10 mg (95%). Orange solid. ^1^H NMR (DMSO-*d*_6_, δ, ppm, J/Hz): δ 8.64–8.62 (m, 1H), 8.52–8.42 (m, 3H), 8.40–8.34 (m, 2H), 8.34 (d, *J =* 8.0 Hz, 1H), 8.30–8.21 (m, 1H), 8.04–7.89 (m, 1H), 7.73–7.66 (m, 1H), 7.65–7.60 (m, 1H), 7.48–7.43 (m, 1H), 5.84–5.81 (m, 1H), 5.48 (d, *J =* 5.3 Hz, 1H), 5.37–5.30 (m, 1H), 5.11–5.03 (m, 1H), 4.17–4.13 (m, 1H), 4.10–4.06 (m, 1H), 3.95–3.92 (m, 1H), 3.81–3.76 (m, 1H), 3.69–3.64 (m, 1H), 3.06–3.01 (m, 2H), 1.73–1.67 (m, 2H), 1.44–1.38 (m, 2H), 1.35–1.32 (m, 2H), 1.29–1.20 (m, 14H), 0.85 (t, *J =* 7.1 Hz, 3H). ^13^C NMR (DMSO-*d*_6_, δ, ppm): δ 161.4, 154.0, 149.6, 143.8, 139.3, 133.7, 132.4, 131.4, 131.2, 130.6, 130.5, 130.3, 127.9, 127.5, 127.1, 126.9, 126.8, 126.8, 125.6, 124.6, 121.5, 121.3, 119.7, 98.4, 92.9, 91.6, 88.7, 88.5, 84.7, 84.1, 74.9, 74.0, 69.1, 68.1, 60.1, 59.5, 32.5, 31.2, 30.2, 29.0, 28.9, 28.8, 28.6, 22.0, 13.9. HRMS *m*/*z* = 685.3279 [M − H]^−^. Calculated *m*/*z* = 685.3272 (C_43_H_45_N_2_O_6_^−^).

**5-[9(10)-Dodecylperylen-3-ylethynyl]-*deoxy*-uridine (11c).** Yield 12 mg (95%). Orange solid. ^1^H NMR (DMSO-*d*_6_, δ, ppm, J/Hz): δ 8.52 (d, *J =* 4.0 Hz, 1H), 8.49–8.41 (m, 2H), 8.38–8.31 (m, 2H), 8.29–8.20 (m, 1H), 8.04–7.96 (m, 1H), 7.93–7.88 (m, 1H), 7.72–7.59 (m, 2H), 7.50–7.42 (m, 1H), 7.29–7.26 (m, 1H), 6.17 (t, 1H), 5.28–5.25 (m, 1H), 5.18–5.14 (m, 1H), 4.33–4.27 (m, 1H), 4.25–4.17 (m, 1H), 3.86–3.83 (m, 2H), 3.75–3.61 (m, 2H), 3.05–3.00 (m, 2H), 1.71–1.67 (m, 2H), 1.43–1.37 (m, 2H), 1.34–1.32 (m, 2H), 1.27–1.21 (m, 14H), 0.89–0.81 (m, 3H). ^13^C NMR (DMSO-*d*_6_, δ, ppm): δ 171.0, 161.5, 153.8, 153.0, 149.4, 139.9, 139.4, 138.5, 137.2, 133.7, 132.4, 132.4, 131.4, 130.9, 130.5, 128.4, 128.0, 127.5, 127.2, 127.1, 126.8, 124.8, 124.7, 124.2, 121.6, 121.5, 121.3, 121.2, 120.8, 120.2, 119.7, 118.6, 106.7, 104.1, 98.4, 90.7, 88.3, 87.6, 85.0, 69.7, 60.8, 32.5, 32.4, 31.2, 30.9, 30.2, 30.2, 29.0, 28.9, 28.8, 28.6, 22.0, 22.0, 13.9. HRMS *m*/*z* = 671.3491 [M + H]^+^. Calculated *m*/*z* = 671.3479 (C_43_H_47_N_2_O_5_^+^).


**General procedure for the synthesis of a mixture of 4-[9(10)-alkylperylen-3-ylethynyl]phenols (12a–c), 3-[9(10)-alkylperylen-3-ylethynyl]phenols (13a–c), and 2-bromo-4-[9(10)-alkylperylen-3-ylethynyl]phenols (14a–c):**


The corresponding iodophenol (1 equiv) and 3-ethynyl-9(10)-alkylperylene **5a–c** (1.3 equiv) were dissolved in dry DMF (3 mL/25 mg starting ethynyl compound) and the mixture was evacuated and flushed with argon five times to remove traces of oxygen. Then, tetrakis(triphenylphosphine)palladium(0) (0.1 equiv) and triethylamine (5 equiv) were added, the mixture was evacuated and refilled with argon 3 times, after which copper iodide (0.5 eq.) was added, and the mixture was evacuated and refilled with argon 3 times. The Schlenk flask was then placed in a glycerol bath at 80 °C and the mixture was stirred for 3 h in the dark. After the almost complete disappearance of the starting product, the mixture was poured into 2.5% (*w*/*w*, aq.) disodium EDTA. The mixture was extracted with 3 portions of ethyl acetate. The organic layer was thoroughly washed with disodium EDTA, water (4 times), and brine and dried over anhydrous Na_2_SO_4_. The solvent was removed in vacuo. The residue was purified by column chromatography on silica gel with a gradient of EtOAc in toluene (0→1%, *v*/*v*) to obtain the desired product as an orange-yellow (**12a–c**, **13a–c**) or black-red (**14a–c**) solid.

**4-[9(10)-Butylperylen-3-ylethynyl]phenol (12a)** was prepared from 25 mg (75.2 µmol) of **5a**; yield 15 mg (48%). Orange solid. ^1^H NMR (CDCl_3_, δ, ppm, J/Hz): δ 8.30–8.21 (m, 3H), 8.17–8.11 (m, 2H), 7.95–7.89 (m, 1H), 7.74–7.68 (m, 1H), 7.63–7.57 (m, 1H), 7.57–7.51 (m, 3H), 7.40–7.34 (m, 1H), 6.89–6.85 (m, 2H), 5.30 (c, 1H), 3.04 (t, *J =* 7.9 Hz, 2H), 1.78–1.74 (m, 2H), 1.51–1.45 (m, 2H), 1.00 (t, *J =* 7.4 Hz, 3H). ^13^C NMR (CDCl_3_, δ, ppm): δ 156.0, 139.8, 139.5, 134.7, 133.5, 133.5, 133.2, 132.0, 131.9, 131.8, 131.6, 131.5, 131.0, 130.9, 130.2, 129.4, 129.1, 129.0, 128.5, 128.5, 127.4, 127.3, 127.1, 127.0, 126.5, 126.4, 126.2, 125.8, 124.5, 124.2, 120.9, 120.8, 120.8, 120.6, 120.6, 120.3, 120.2, 119.8, 119.3, 116.1, 116.1, 115.8, 115.6, 95.4, 95.3, 87.0, 36.7, 33.7, 33.6, 33.3, 33.2, 32.9, 32.1, 31.7, 29.9, 29.5, 24.9, 23.0, 22.8, 22.8, 14.3, 14.2. HRMS m/z = 423.1752 [M − H]^−^. Calculated *m*/*z* = 423.1744 (C_32_H_23_O^−^).

**3-[9(10)-Butylperylen-3-ylethynyl]phenol (13a)** was prepared from 20 mg (45.0 µmol) of **5c**; yield 16 mg (50%). Orange solid. ^1^H NMR (CDCl_3_, δ, ppm, J/Hz): δ 8.29–8.21 (m, 3H), 8.17–8.13 (m, 1H), 8.14–8.11 (m, 1H), 7.94–7.90 (m, 1H), 7.75–7.71 (m, 1H), 7.62–7.56 (m, 1H), 7.56–7.51 (m, 1H), 7.36 (d, *J =* 7.6 Hz, 1H), 7.30–7.26 (m, 1H), 7.26–7.22 (m, 1H), 7.13 (c, 1H), 6.86 (d, *J =* 8.0 Hz, 1H), 4.78 (c, 1H), 3.04 (t, *J =* 7.9 Hz, 2H), 1.79–1.70 (m, 2H), 1.52–1.45 (m, 2H), 1.00 (t, *J =* 7.4 Hz, 3H). ^13^C NMR (CDCl_3_, δ, ppm): δ 155.5, 140.0, 139.6, 134.8, 133.2, 132.5, 132.4, 132.0, 132.0, 131.7, 131.4, 131.3, 131.2, 129.9, 129.3, 129.0, 129.0, 129.0, 128.5, 127.5, 127.4, 127.1, 127.0, 126.5, 126.4, 126.1, 125.7, 124.9, 124.9, 124.6, 124.6, 124.6, 124.3, 121.0, 121.0, 120.8, 120.7, 120.7, 120.4, 120.1, 119.7, 119.6, 119.3, 118.4, 118.4, 116.0, 116.0, 95.0, 94.9, 88.4, 88.4, 33.3, 33.2, 33.0, 23.0, 22.8, 14.3, 14.2. HRMS *m*/*z* = 425.1896 [M + H]^+^. Calculated *m*/*z* = 425.1900 (C_32_H_25_O^+^).

**2-Bromo-4-[9(10)-butylperylen-3-ylethynyl]phenol (14a)** was prepared from 20 mg (45.0 µmol) of **5c**; yield 14 mg (53%). Dark red solid. ^1^H NMR (CDCl_3_, δ, ppm, J/Hz): δ 8.25–8.17 (m, 3H), 8.16–8.08 (m, 2H), 7.94–7.89 (m, 1H), 7.80–7.77 (m, 1H), 7.71–7.67 (m, 1H), 7.60–7.54 (m, 1H), 7.54–7.46 (m, 2H), 7.37–7.33 (m, 1H), 7.05 (d, *J =* 8.3 Hz, 1H), 5.68 (c, 1H), 3.03 (t, *J =* 7.9 Hz, 2H), 1.79–1.72 (m, 2H), 1.52–1.45 (m, 2H), 1.00 (t, *J =* 7.4 Hz, 3H). ^13^C NMR (CDCl_3_, δ, ppm): δ 152.7, 152.7, 140.0, 139.6, 135.1, 135.1, 134.7, 133.1, 132.9, 132.8, 132.4, 132.3, 132.0, 132.0, 131.7, 131.4, 131.2, 131.1, 130.2, 129.3, 129.0, 129.0, 129.0, 128.5, 127.5, 127.4, 127.1, 127.0, 126.5, 126.4, 126.0, 125.6, 124.6, 124.3, 121.0, 120.9, 120.8, 120.7, 120.7, 120.3, 120.0, 119.7, 119.6, 119.3, 117.5, 117.4, 116.3, 110.2, 93.7, 93.6, 88.0, 88.0, 33.3, 33.2, 32.9, 32.1, 29.9, 29.8, 29.5, 23.0, 22.8, 14.3, 14.2. HRMS *m*/*z* = 502.0955 [M^+^]. Calculated *m*/*z* = 502.0932 (C_32_H_23_BrO^+^).

**4-[9(10)-Octylperylen-3-ylethynyl]phenol (12b)** was prepared from 25 mg (64.3 µmol) of **5b**; yield 16 mg (51%). Orange solid. ^1^H NMR (CDCl_3_, δ, ppm, J/Hz): δ 8.31–8.20 (m, 3H), 8.17–8.10 (m, 2H), 7.93–7.89 (m, 1H), 7.73–7.69 (m, 1H), 7.61–7.55 (m, 1H), 7.57–7.51 (m, 3H), 7.38–7.34 (m, 1H), 6.87 (d, *J =* 8.0 Hz, 2H), 4.94 (c, 1H), 3.03 (t, 2H), 1.80–1.73 (m, 2H), 1.48–1.44 (m, 2H), 1.40–1.35 (m, 2H), 1.35–1.26 (m, 6H), 0.91–0.87 (m, 3H). ^13^C NMR (CDCl_3_, δ, ppm): δ 156.0, 155.9, 139.8, 139.5, 134.7, 133.5, 133.5, 133.1, 132.0, 132.0, 131.9, 131.7, 131.5, 131.0, 130.9, 129.3, 129.1, 129.0, 128.5, 127.3, 127.3, 127.0, 127.0, 126.5, 126.4, 126.2, 125.8, 124.5, 124.2, 120.9, 120.8, 120.8, 120.6, 120.6, 120.3, 120.1, 119.8, 119.3, 116.1, 116.1, 115.8, 95.4, 95.3, 87.0, 87.0, 33.6, 33.6, 32.1, 30.8, 30.0, 29.9, 29.8, 29.7, 29.5, 29.5, 22.8, 14.3. HRMS *m*/*z* = 480.2456 [M^+^]. Calculated *m*/*z* = 480.2448 (C_36_H_32_O^+^).

**3-[9(10)-Octylperylen-3-ylethynyl]phenol (13b)** was prepared from 25 mg (64.3 µmol) of **5b**; yield 17 mg (53%). Orange solid. ^1^H NMR (CDCl_3_, δ, ppm, J/Hz): δ 8.29–8.20 (m, 3H), 8.17–8.10 (m, 2H), 7.94–7.89 (m, 1H), 7.75–7.70 (m, 1H), 7.61–7.56 (m, 1H), 7.56–7.51 (m, 1H), 7.38–7.34 (m, 1H), 7.30–7.26 (m, 1H), 7.26–7.22 (m, 1H), 7.12 (c, 1H), 6.88–6.84 (m, 1H), 4.78 (c, 1H), 3.03 (t, *J =* 7.9 Hz, 2H), 1.77 (*p*, *J =* 7.7 Hz, 2H), 1.46 (t, *J =* 7.6 Hz, 2H), 1.41–1.35 (m, 2H), 1.37–1.25 (m, 6H), 0.89 (t, *J =* 6.9 Hz, 3H). ^13^C NMR (CDCl_3_, δ, ppm): δ 155.5, 140.0, 139.6, 134.8, 133.1, 132.5, 132.4, 132.0, 132.0, 131.7, 131.4, 131.3, 131.2, 129.9, 129.3, 129.0, 129.0, 129.0, 128.5, 127.5, 127.4, 127.1, 127.0, 126.5, 126.4, 126.1, 125.7, 124.9, 124.9, 124.6, 124.6, 124.6, 124.3, 121.0, 121.0, 120.8, 120.7, 120.7, 120.3, 120.1, 119.7, 119.6, 119.3, 118.4, 118.4, 116.0, 116.0, 95.0, 94.9, 88.4, 88.4, 77.3, 77.2, 77.0, 33.6, 33.6, 32.1, 30.8, 30.0, 29.9, 29.7, 29.5, 22.8, 14.3. HRMS *m*/*z* = 480.2448 [M^+^]. Calculated *m*/*z* = 480.2488 (C_36_H_32_O^+^).

**2-Bromo-4-[9(10)-octylperylen-3-ylethynyl]phenol (14b)** was prepared from 25 mg (64.3 µmol) of **5b**; yield 18 mg (56%). Dark red solid. ^1^H NMR (CDCl_3_, δ, ppm, J/Hz): δ 8.25–8.18 (m, 3H), 8.15–8.07 (m, 2H), 7.92–7.88 (m, 1H), 7.79–7.76 (m, 1H), 7.71–7.66 (m, 1H), 7.60–7.55 (m, 1H), 7.54–7.48 (m, 2H), 7.36–7.32 (m, 1H), 7.05 (d, *J =* 8.3 Hz, 1H), 5.67 (c, 1H), 3.01 (t, *J =* 7.9 Hz, 2H), 1.80–1.73 (m, 2H), 1.49–1.43 (m, 2H), 1.40–1.34 (m, 2H), 1.34–1.24 (m, 6H), 0.89 (t, *J =* 7.0 Hz, 3H). ^13^C NMR (CDCl_3_, δ, ppm): δ 152.7, 152.7, 140.0, 139.6, 135.1, 135.1, 134.7, 133.1, 132.9, 132.8, 132.4, 132.3, 132.0, 132.0, 131.7, 131.4, 131.2, 131.1, 129.3, 129.0, 129.0, 129.0, 128.5, 127.5, 127.4, 127.1, 127.0, 126.5, 126.4, 126.0, 125.6, 124.6, 124.3, 121.0, 120.9, 120.8, 120.7, 120.7, 120.3, 120.0, 119.7, 119.6, 119.3, 117.5, 117.4, 116.3, 110.2, 93.7, 93.6, 88.0, 88.0, 33.6, 33.6, 32.1, 30.8, 30.0, 29.9, 29.7, 29.5, 29.5, 22.8, 14.3. HRMS *m*/*z* = 559.1562 [M^+^]. Calculated *m*/*z* = 559.1553 (C_36_H_31_BrO^+^).

**4-[9(10)-Dodecylperylen-3-ylethynyl]phenol (12c)** was prepared from 25 mg (56.2 µmol) of **5c**; yield 15 mg (53%). Orange solid. ^1^H NMR (CDCl_3_, δ, ppm, J/Hz): δ 8.25 (td, *J =* 22.2, 21.7, 7.7 Hz, 3H), 8.17–8.10 (m, 2H), 7.93–7.89 (m, 1H), 7.73–7.69 (m, 1H), 7.61–7.55 (m, 1H), 7.57–7.51 (m, 3H), 7.36 (d, *J =* 6.8 Hz, 1H), 6.87 (d, *J =* 7.2 Hz, 2H), 4.93 (c, 1H), 3.03 (t, *J =* 7.7 Hz, 2H), 1.80–1.73 (m, 2H), 1.48–1.43 (m, 2H), 1.41–1.32 (m, 2H), 1.31–1.25 (m, 14H), 0.91–0.86 (m, 3H). ^13^C NMR (CDCl_3_, δ, ppm): δ 156.0, 155.9, 139.8, 139.5, 134.7, 133.5, 133.5, 133.1, 132.0, 132.0, 131.9, 131.7, 131.5, 131.0, 130.9, 130.2, 129.3, 129.1, 129.0, 128.5, 128.5, 127.3, 127.3, 127.0, 127.0, 126.5, 126.4, 126.2, 125.8, 124.5, 124.2, 120.9, 120.8, 120.8, 120.6, 120.6, 120.3, 120.1, 119.7, 119.3, 116.1, 116.1, 115.8, 115.6, 95.4, 95.3, 87.0, 87.0, 33.9, 33.6, 33.5, 32.1, 30.8, 30.0, 29.8, 29.8, 29.8, 29.7, 29.5, 22.8, 14.3. HRMS *m*/*z* = 537.3153 [M + H]^+^. Calculated *m*/*z* = 537.3152 (C_40_H_41_O^+^).

**3-[9(10)-Dodecylperylen-3-ylethynyl]phenol (13c)** was prepared from 25 mg (56.2 µmol) of **5c**; yield 16 mg (54%). Orange solid. ^1^H NMR (CDCl_3_, δ, ppm, J/Hz): δ 8.29–8.20 (m, 3H), 8.17–8.09 (m, 2H), 7.94–7.89 (m, 1H), 7.75–7.71 (m, 1H), 7.62–7.56 (m, 1H), 7.56–7.51 (m, 1H), 7.38–7.34 (m, 1H), 7.30–7.26 (m, 1H), 7.26–7.22 (m, 1H), 7.14–7.11 (m, 1H), 6.88–6.84 (m, 1H), 4.80 (c, 1H), 3.03 (t, *J =* 7.7 Hz, 2H), 1.77 (*p*, *J =* 7.7 Hz, 2H), 1.48–1.41 (m, 2H), 1.40–1.34 (m, 2H), 1.32–1.24 (m, 14H), 0.88 (t, *J =* 7.0 Hz, 3H). ^13^C NMR (CDCl_3_, δ, ppm): δ 155.5, 140.1, 139.6, 134.8, 133.1, 132.5, 132.4, 132.0, 132.0, 131.7, 131.4, 131.3, 131.2, 129.9, 129.3, 129.0, 129.0, 129.0, 128.5, 127.5, 127.4, 127.1, 127.0, 126.5, 126.4, 126.1, 125.7, 124.9, 124.9, 124.6, 124.6, 124.6, 124.3, 121.0, 121.0, 120.8, 120.7, 120.7, 120.3, 120.1, 119.7, 119.6, 119.3, 118.4, 118.4, 116.0, 116.0, 95.0, 94.9, 88.4, 88.4, 33.6, 33.6, 32.1, 31.6, 30.8, 30.5, 30.4, 30.0, 29.8, 29.8, 29.7, 29.5, 22.8, 14.3. HRMS *m*/*z* = 536.3084 [M^+^]. Calculated *m*/*z* = 536.3074 (C_40_H_40_O^+^).

**2-Bromo-4-[9(10)-dodecylperylen-3-ylethynyl]phenol (14c)** was prepared from 25 mg (56.2 µmol) of **5c**; yield 17 mg (58%). Dark red solid. ^1^H NMR (CDCl_3_, δ, ppm, J/Hz): δ 8.25–8.18 (m, 3H), 8.15–8.07 (m, 2H), 7.92–7.88 (m, 1H), 7.79–7.76 (m, 1H), 7.71–7.66 (m, 1H), 7.60–7.55 (m, 1H), 7.55–7.48 (m, 2H), 7.36–7.32 (m, 1H), 7.05 (d, *J =* 8.3 Hz, 1H), 5.68 (c, 1H), 3.02 (t, *J =* 7.9 Hz, 2H), 1.79–1.72 (m, 2H), 1.49–1.43 (m, 2H), 1.40–1.34 (m, 2H), 1.33–1.25 (m, 14H), 0.88 (t, *J =* 7.1 Hz, 3H). ^13^C NMR (CDCl_3_, δ, ppm): δ 152.7, 152.7, 140.0, 139.6, 135.1, 135.1, 134.6, 133.1, 132.9, 132.8, 132.4, 132.3, 132.0, 132.0, 131.7, 131.4, 131.2, 131.1, 130.2, 129.3, 129.0, 129.0, 128.9, 128.5, 127.5, 127.4, 127.0, 127.0, 126.5, 126.4, 126.0, 125.6, 124.6, 124.3, 121.0, 120.9, 120.8, 120.7, 120.6, 120.3, 120.0, 119.7, 119.5, 119.2, 117.5, 117.4, 116.3, 110.2, 93.7, 93.6, 88.0, 88.0, 33.6, 33.5, 32.1, 30.8, 30.0, 29.8, 29.8, 29.7, 29.5, 22.8, 14.3. HRMS *m*/*z* = 613.2108 [M^+^]. Calculated *m*/*z* = 613.2101 (C_40_H_38_BrO^+^).


**General procedure for the synthesis of a mixture of 5-[9(10)-alkylperylen-3-yl]thiophene-2-carboxylic acids (15a–c):**


To 3-chloro-3-[9(10)-alkylperpylen-3-yl]acrolein **4a–c** (1 equiv) was added isopropanol (1 mL per 4 mg of starting material) and KOH (5 equiv). The mixture was refluxed with stirring for 72 h. Then, 30 mL of 10% aqueous HCl was added, boiled for 1 h with stirring, and the mixture was filtered through a fritted glass filter (16–40 μm porosity). The precipitate was washed with 3 portions of water, 2 portions of methylene chloride, and 1 portion of ethyl acetate yielding an orange-brown powder.

**5-[9(10)-Butylperylen-3-yl]thiophene-2-carboxylic acid (15a)** was prepared from 20 mg (50.4 µmol) of **4a**; yield 12 mg (49%). Orange solid. ^1^H NMR (DMSO-*d*_6_, δ, ppm, J/Hz): δ 8.46–8.39 (m, 2H), 8.38–8.31 (m, 2H), 8.15–8.09 (m, 1H), 8.00–7.95 (m, 1H), 7.64–7.56 (m, 3H), 7.48–7.46 (m, 1H), 7.49–7.43 (m, 1H), 7.28–7.25 (m, 1H), 3.04 (t, *J =* 7.9 Hz, 2H), 1.73–1.67 (m, 2H), 1.49–1.42 (m, 2H), 0.97 (t, *J =* 7.4 Hz, 3H). ^13^C NMR (DMSO-*d*_6_, δ, ppm): δ 172.4, 163.2, 139.1, 139.0, 132.4, 132.0, 131.1, 130.9, 128.7, 128.3, 128.1, 128.0, 127.7, 127.5, 127.1, 126.8, 125.1, 124.7, 124.1, 121.1, 121.0, 120.9, 120.8, 120.6, 120.3, 119.8, 32.4, 32.2, 32.1, 22.2, 13.8. HRMS *m*/*z* = 433.1265 [M^—^H]^–^. Calculated *m*/*z* = 433.1257 (C_29_H_21_O_2_S^–^).

**5-[9(10)-Octylperylen-3-yl]thiophene-2-carboxylic acid (15b)** was prepared from 20 mg (44.1 µmol) of **4b**; yield 14 mg (64%). Orange solid. ^1^H NMR (DMSO-*d*_6_, δ, ppm, J/Hz): δ 8.47–8.38 (m, 2H), 8.38–8.31 (m, 2H), 8.06–8.00 (m, 1H), 8.00–7.95 (m, 1H), 7.82 (d, *J =* 3.7 Hz, 1H), 7.66–7.58 (m, 3H), 7.45–7.40 (m, 2H), 3.01 (t, *J =* 7.8 Hz, 2H), 1.73–1.65 (m, 2H), 1.45–1.37 (m, 2H), 1.37–1.30 (m, 2H), 1.32–1.20 (m, 6H), 0.85 (t, *J =* 7.0 Hz, 3H). ^13^C NMR (DMSO-*d*_6_, δ, ppm, J/Hz): δ 172.0, 162.8, 139.6, 139.2, 133.5, 132.4, 131.9, 131.8, 131.3, 131.2, 130.8, 130.4, 129.8, 129.2, 129.1, 128.7, 128.7, 128.4, 128.1, 128.1, 128.0, 127.9, 127.8, 127.2, 127.1, 126.9, 126.8, 124.7, 124.6, 124.3, 121.2, 121.2, 120.7, 120.2, 119.7, 40.1, 32.5, 32.4, 31.3, 30.2, 29.1, 28.9, 28.7, 22.1, 21.0, 13.9. HRMS *m*/*z* = 491.2037 [M + H]^+^. Calculated *m*/*z* = 491.2039 (C_33_H_31_O_2_S^+^).

**5-[9(10)-Dodecylperylen-3-yl]thiophene-2-carboxylic acid (15c)** was prepared from 20 mg (39.3 µmol) of **4c**; yield 11 mg (58%). Orange solid. ^1^H NMR (DMSO-d6, δ, ppm, J/Hz): δ 8.49–8.41 (m, 2H), 8.41–8.34 (m, 2H), 8.06–7.97 (m, 2H), 7.87–7.82 (m, 1H), 7.68–7.60 (m, 3H), 7.47–7.43 (m, 2H), 3.03 (t, *J =* 7.8 Hz, 2H), 1.72–1.68 (m, 2H), 1.45–1.38 (m, 2H), 1.36–1.31 (m, 2H), 1.29–1.20 (m, 14H), 0.88–0.81 (m, 3H). ^13^C NMR (DMSO-*d*_6_, δ, ppm): it was not possible to obtain a ^13^C spectrum of the compound due to too low solubility and a large number of quaternary carbon atoms. (^1^H–^13^C) HMBC: ^13^C NMR (DMSO-*d*_6_, δ, ppm): δ 128.6, 124.7, 131.8, 128.6, 140.1, 128.7, 162.9, 128.5, 121.6, 128.6, 140.1, 131.8, 132.9, 128.7, 132.9, 33.0, 30.6, 127.7, 29.4, 30.6, 140.1, 132.9, 127.7, 30.6, 40.2, 40.9, 40.2, 29.4, 33.0, 31.7, 29.4, 27.6, 22.5, 31.7; ^1^H NMR (DMSO-*d*_6_, δ, ppm): δ 8.5, 8.4, 8.4, 8.4, 8.3, 8.1, 8.0, 8.0, 8.0, 8.0, 8.0, 7.6, 7.6, 7.5, 7.5, 7.4, 3.0, 3.0, 3.0, 3.0, 3.0, 3.0, 3.0, 3.0, 2.6, 2.6, 2.4, 1.7, 1.7, 1.3, 1.2, 1.2, 0.8, 0.8. (^1^H–^13^C) HSQC: ^13^C NMR (DMSO-*d*_6_, δ, ppm): δ 121.6, 120.7, 121.2, 120.3, 121.5, 125.4, 125.0, 124.7, 129.4, 127.3, 128.2, 127.6, 128.7, 55.4, 70.3, 0.1, 33.0, 40.9, 40.8, 30.7, 29.5, 29.4, 22.6, 29.5, 29.2, 31.8, 14.5; ^1^H NMR (DMSO-*d*_6_, δ, ppm): δ 8.4, 8.4, 8.4, 8.4, 8.3, 8.1, 8.1, 8.0, 7.6, 7.6, 7.6, 7.4, 7.4, 5.7, 3.5, 3.3, 3.0, 2.5, 2.5, 1.7, 1.4, 1.3, 1.2, 1.2, 1.2, 1.2, 0.8. HRMS *m*/*z* = 545.2518 [M − H]^−^. Calculated *m*/*z* = 545.2509 (C_37_H_37_O_2_S^−^).

### 3.1. Spectral Properties

Samples of the compounds were dissolved in DMSO to obtain 4 μm stock solutions. 5 μL of these solutions were diluted 200 times in MeOH (995 μL) and absorption spectra were recorded.

The solutions used to record the absorption spectra were diluted 200 times in methanol and fluorescence emission spectra were recorded with an excitation wavelength of 440 nm. Fluorescence excitation spectra were recorded with detection at the fluorescence emission maxima.

Data were processed in Excel and GraphPad Prism 8 (normalization and plotting).

### 3.2. Solubilization in 15% DMSO

A 100 µL amount of 4 µM stock solutions of the compounds were diluted with 560 µL of water to give a concentration of 15% DMSO in water. The mixtures were then centrifuged at 20,000 rcf for 5 min; 500 μL of the supernatant were collected and lyophilized; 76 µL of DMSO were added to the dry residue, and the absorbance was measured at the maximum absorption wavelength for compounds in solutions diluted 200 times with methanol. The data obtained were recalculated according to the law of dilutions and Beer’s law to obtain the initial concentrations of the compounds solubilized in 15% DMSO in water.

### 3.3. Quantum Yields of Singlet Oxygen Photogeneration

Quantum yields of singlet oxygen photogeneration were measured as reported [23,24] in methanol solution using DPBF (Sigma-Aldrich, Darmstadt, Germany) as the singlet oxygen scavenger. Spectrophotometric measurements were performed in a Qpod 2e thermostated cuvette holder (Quantum Northwest, Liberty Lake, WA, USA) at 25 °C and with magnetic stirring (500 rpm). Absorption spectra were recorded using a MayaPro spectrophotometer (Ocean Optics, Orlando, FL, USA) and a stabilized white light source with an SLS201L tungsten lamp (Thorlabs, Newton, NJ, USA).

To study photosensitized ^1^O_2_ generation, we used a white MCWHLP1 LED (Thorlabs, Newton, NJ, USA) with filters to limit the radiation to the 450–470 nm range (5.5 mW/cm^2^). Illumination was uniform over the entire volume of the cuvette, to prevent artifacts associated with the diffusion of non-reacted components into the illuminated volume of the cuvette. Illumination was performed in pulsed mode, with 1 s of illumination followed by 5 s of dark adaptation, during which the absorption spectrum of the photosensitizer-DPBF solution was recorded.

Singlet oxygen generation quantum yield was calculated according to Equation (1):(1)$$\varphi_{\Delta }=\varphi_{\Delta }^{0}*\frac{r}{r^{0}}\frac{\left(1-10^{{-A}^{0}}\right)}{\left(1-10^{-A}\right)}$$
where r is the rate of DPBF bleaching in solution of the photosensitizer (PS), A is the PS absorbance in the region of illumination, and index 0 represents reference PS (we used riboflavin with $\varphi_{\Delta }^{0}=0.51$ in methanol [73]. The data processing procedure was performed as described earlier [74].

### 3.4. Molecular Dynamics Simulations

To study the embedding of the perylene-related compounds into lipid bilayers, a set of MD simulations of compounds **12** and **14** with alkyl chains of various lengths (C_0_, C_4_, C_8_, C_12_) was carried out in an explicit membrane/water environment. The CHARMM-GUI v3.7 software package [75] was used to predict CHARMM36 forcefield parameters for compounds **12**, **14**, **15** and their alkyl derivatives using the following instruments: Ligand Reader and Modeler [76], PDB Reader and Manipulator [77], and Force Field Converter [78]. The predicted parameters for all atoms showed penalties below 30, which indicates that the forcefield parameters can be considered reliable. A zwitterionic 1-palmitoyl-2-oleoyl-*sn*-phosphatidylcholine (POPC) bilayer was used. The perylene derivatives were randomly placed in an aqueous medium above a pre-equilibrated bilayer (64 lipid molecules per monolayer) with a 1 nm minimum distance to the lipids using an in-house software framework written in C++ and Python.

MD simulations were carried out using the GROMACS package [79] version 2020.6 and the CHARMM36 all-atom force field [80,81,82]. In calculations, the tip3p [83] water model, 2 fs integration time step, and imposed 3D periodic boundary conditions were employed. A spherical cutoff function (1.2 nm) and the particle mesh Ewald (PME) algorithm (with a 1.2 nm cutoff) [84] were used to process van der Waals and electrostatic interactions, respectively. MD production runs were conducted in an NPT ensemble at a constant temperature of 310 K and a semi-isotropic pressure of 1 bar maintained using the V-rescale [85] and Parrinello–Rahman algorithms [86] with 1.0 and 0.1 ps relaxation parameters, respectively. Before the MD production runs, all systems were first equilibrated using 5000 steps of steepest descent minimization, followed by heating from 5 to 310 K over 0.5 ns MD-run. Small molecules, lipids, and solvent molecules were coupled separately. MD simulation of each system was repeated three times with random assignment of initial velocities. Trajectory length ranged from 200 to 400 ns. The total MD simulation time in this work is equal to 8 µs.

System configurations extracted from MD trajectories were centered on the perylene group and sampled for analysis at time intervals of 10–100 ps using original GROMACS utilities. All MD trajectories were analyzed starting from 100 ns simulation time when all compounds were embedded into the bilayer. Density profiles of compounds, rotational angles, and COM coordinates for the molecules were calculated using *gmx density*, *gmx angle*, and *gmx traj* utilities, respectively. The accessible surface area of membrane-embedded perylene molecules was estimated by Naccess software v. 2.1.1 [87]. In-house Python scripts that use NumPy and Matplotlib libraries were used for plotting. Molecular graphics were rendered using PyMOL v. 2.5.0 (Schrödinger, Inc., New York, NY, USA, http://pymol.org (accessed on 16 November 2023)) [88].

### 3.5. Biological Studies

#### 3.5.1. Viruses and Cells

HSV-1, strain MacIntyre, kindly provided by Prof. Andreas Sauerbrei, German Reference Laboratory for HSV and VZV, Germany, was used for our antiviral studies. Vero cells (ATCC CCL-81, African Green Monkey, adult kidney, epithelial) were used for HSV-1 propagation, anti-HSV-1 assays, and HSV-1-based plaque assays. Vero cells were cultured in Dulbecco’s Modified Eagle’s Medium (DMEM) supplemented with 10% newborn calf serum, 100 U/mL penicillin, 100 µg/mL streptomycin, and 1% glutamine (Sigma-Aldrich, Prague, Czech Republic). Vero cells were cultured at 37 °C under 5% CO_2_.

#### 3.5.2. Cytotoxicity Assay

Vero cells were cultured for 24 h in 96-well plates to form a confluent monolayer and then were treated with the tested compounds at concentrations of 50 µM. After 48 h of cultivation in the dark at 37 °C under 5% CO_2_, the cell culture medium was aspirated. The potential cytotoxicity of the tested derivatives was determined based on cell viability using Cell Counting Kit-8 (Dojindo Molecular Technologies, Munich, Germany) according to the manufacturer’s instructions.

HEK-293T cells were seeded on 96-well plates a day before treatment in the amount of 5 × 10^4^ cells per well. The next day, the cells were treated with the compounds at concentrations of 10 µM and cultured in the dark in standard culturing conditions. Cell viability was assessed after 48 h of cultivation with the CellTiter-GLO 2.0 cell viability assay (Promega, Madison, WI, USA).

#### 3.5.3. VSV Inhibition Analysis

For VSV stock preparation, 15 cm dishes containing monolayer culture of HEK-293T cells were infected with VSV Indiana strain at MOI = 0.1 in 5 mL of DMEM/F12 (PAA) without serum, cells were placed in a 37 °C incubator for virus adsorption for 1 h; then, media was replaced with 18 mL of DMEM/F12 high glucose with 2% Fetal Bovine Serum (FBS) (Gibco/Thermo Fisher Scientific, Waltham, MA, USA). After 3 days of incubation when all cells were detached, virus-containing supernatants were collected, centrifuged for 15 min at 3000× *g*, and filtered through 0.45 μm syringe filters. To assess the antiviral effects on VSV titers, the compounds were dissolved in DMSO at concentrations starting from 2 mM with 2-fold serial dilutions. Then, DMSO/PBS working solutions were prepared, containing 17.5% of each DMSO stock in PBS for EC_50_ measurements starting from 50 µM or 3.5% of each DMSO stock in PBS for EC_50_ measurements starting from 10 µM final concentrations. The control working solution consisted of 17.5% or 3.5% DMSO in PBS. To measure antiviral effects at the 200 μM concentration, 10% of 2 mM DMSO stock was added to the virus. Otherwise, 30 μL aliquots of VSV stock were mixed with 5 μL of each working solution, bringing the final DMSO concentration to 2.5% or 0.5%, respectively, and left under fluorescent light for 1 h for EC_50_ determination.

For TCID_50_ measurements, HEK-293T cells were seeded the previous day in 96-well plates at densities of 8 × 10^4^ cells per well. On the day of the experiment, cells were infected with 40 μL of serial 2-fold dilutions of virus samples in DMEM/F12 media for 1 h. Each dilution was carried out in 6 repetitions. After the initial infection, the media was replaced with 100 µL of fresh DMEM/F12 media supplemented with 2% FBS. After 2 days of incubation, cytopathic effects (CPE) were scored and TCID_50_ values were calculated using the improved Kärber method [89].

#### 3.5.4. HSV-1 Titer Reduction Assay

Vero cells were seeded in 96-well plates and incubated for 24 h to form a confluent monolayer. The virus in DMEM (MOI of 0.01) was mixed with each compound (10 µM) and used for infection of the cells. At 48 h post infection (p.i.), the culture medium was collected and viral titers were determined by plaque assays.

#### 3.5.5. HSV-1 Plaque Assay

To quantify the viral titers for HSV-1, plaque assays were performed using Vero cells. Briefly, 10-fold dilutions of the virus were prepared in 24-well tissue culture plates, and the cells were added to each well (0.6–1.5 × 10^5^ cells/well). After 4 h incubation, the suspension was overlaid with 3% (*w*/*v*) carboxymethylcellulose in DMEM. Following day 4 of incubation at 37 °C and 5% CO_2_, the infected plates were washed with phosphate-buffered saline, and the cell monolayers were stained with naphthalene black. The virus titer was expressed as plaque-forming units (PFU)/mL.

#### 3.5.6. Studies on Photodynamic Inactivation of HSV-1

The virus in DMEM (titer of 10^4^ PFU/mL) was mixed with selected compounds (0–10 µM) in a microtiter plate in daylight and irradiated for 10 min at RT with LEDs (465–480 nm) at an approximate power density of 30 mW/cm^2^. As negative controls, the virus was mixed with selected compounds (0–10 µM) in daylight and incubated with the compound for 10 min in daylight at RT. Subsequently, both irradiated and non-irradiated virus samples were incubated in the dark at 37 °C for an additional 60 min. Viral titers were determined by plaque assays. Note: all manipulation, including sample preparation, pipetting and plaque assays was performed in daylight in both cases (Figure 7A,D). To eliminate the influence of daylight on compound activity, the entire experiment, including all manipulation with the samples, was performed in a dark room under red light (Figure 7H). The virus sample was mixed with selected compounds (0–10 µM), incubated at 37 °C for 60 min, and the viability of the virus was assessed by plaque assays. Plaque assays were also performed under red light.

#### 3.5.7. Studies on Light-Induced Cytotoxicity of Perylene Compounds

To determine light-induced cytotoxicity of the tested compounds, Vero cells were cultured in 96-well plates for 24 h to form a confluent monolayer and then treated with the tested compounds at concentrations ranging from 0 to 10 µM. Subsequently, cells treated with the compounds were irradiated with LEDs (465–480 nm, 30 mW/cm^2^) for 10 min at RT. As a negative control, compound-treated cells were incubated for 10 min in daylight at RT. Then, both the irradiated and non-irradiated cell monolayers were incubated at 37 °C in the dark for 48 h. Subsequently, the cell culture medium was aspirated and the cytotoxicity of the tested perylene derivatives was determined based on cell viability using Cell Counting Kit-8 (Dojindo Molecular Technologies, Munich, Germany) according to the manufacturer’s instructions. Note: all manipulation with samples was performed in daylight (Figure 7K).

#### 3.5.8. HSV-1 Envelope Interaction Studies (Intercalation Assay)

To demonstrate that alkylated perylene compounds interact with the viral envelope, Vero cells were seeded in 6-well plates (approximately 10^6^ cells/well) and incubated for 24 h to form a confluent monolayer. The viral inoculum (10^5^ PFU/mL) was pretreated with compounds (0, 0.08, 1, and 10 µM) for 1 h at 37 °C, diluted to 100 PFU/mL and used to infect Vero cells in 6-well plates for 1 h at 4 °C. Cell monolayers were then washed with PBS to remove un-adsorbed virus, and then fresh medium containing 1.5% carboxymethylcellulose was added to the cells. After 5 days of incubation at 37 °C, the cell monolayers were stained with naphthalene black and the plaque number was determined (Figure 8A).

#### 3.5.9. HSV-1-Based Fusion Assay

To demonstrate that alkylated perylene compounds inhibit the virus–cell fusion process, Vero cells were seeded in 6-well plates (approximately 10^6^ cells/well) and incubated for 24 h to form a confluent monolayer. HSV-1 (100 PFU/mL) was then added to the cells and incubated at 4 °C for 2 h. After incubation, the cell monolayers were washed three times with ice-cold PBS to remove un-adsorbed virus, and fresh ice-cold medium containing the tested compounds (0, 0.08, 2, and 10 µM) was added to the cells and incubated for 2 h at 4 °C. After another 2 h incubation at 37 °C, the medium was aspirated, the cells were washed with PBS and fresh medium containing 1.5% carboxymethylcellulose was added to the cells. After 5 days of incubation at 37 °C, the cell monolayers were stained with naphthalene black and the plaque number was determined (Figure 8B).

#### 3.5.10. Interaction of Perylene with Liposomes

Steady-state fluorescence characteristics of the sample in PBS or in the presence of liposome membrane models were determined using steady-state fluorescence spectroscopy at a constant excitation and emission wavelength, according to the corresponding sample excitation and emission maxima. For the blank, 0.1 mM analyte in DMSO was mixed with 50 μL of PBS. To monitor the interaction of perylene compounds with liposomes, 50 μL of the liposomal suspension were added to the mixture, and fluorescence was measured in L-format using a Chronos DFD Fluorescence spectrometer (ISS, San Antonio, TX, USA) equipped with a 300 W Cermax xenon arc lamp (ISS, San Antonio, TX, USA), a concave holographic grating monochromator and a PMT detector. The required amount of each sample was measured in a 1 cm quartz cuvette, at a constant temperature of 25 °C. The resulting data were evaluated using Vinci software v3 (ISS, San Antonio, TX, USA) and correlated to the utilized optical configuration. Preparation of liposomes was performed as described previously [23].

## 4. Conclusions

In this study, we modified the aromatic moiety of perylene-based antiviral molecules by introducing alkyl substituents of various lengths (C_4_, C_8_, C_12_). The synthesized compounds were found to retain high singlet oxygen generation yields but exhibited a drastic reduction in antiviral activity against enveloped viruses, HSV-1 and VSV. MD simulations revealed that the alkyl tails in the hydrophobic core of the molecules stabilize the perpendicular orientation of the perylene compound within the lipid bilayer. Conversely, non-alkylated compounds with high antiviral activity showed a substantially higher population in a state close to the membrane surface. Therefore, we hypothesize that the photosensitization of the compounds is linked to their localization in the subsurface layer of the membrane, where the perylene group may be accessible to dissolved oxygen in the surrounding water.

## Data Availability

Data are contained within the article and Appendix A.

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
