# Peer review of "Alkyl Derivatives of Perylene Photosensitizing Antivirals: Towards Understanding the Influence of Lipophilicity"

_ijms, 2023, doi:10.3390/ijms242216483_

Round 1

Reviewer 1 Report

Comments and Suggestions for Authors

Dear Authors,

I kindly ask that the following issues with this manuscript be addressed and corrected.

Please ensure that all abbreviations are explained when first used in the text, ex. EBOV, MARV, ZIKV, LASV, etc.,

The presentation of the results of antiviral assays needs to be significantly improved. You wrote that “Antiviral activity is shown in Table 2 and on Figure 6.” (2.4. Antiviral properties). Table 2, entitled “Antiviral activity of synthesized (lipophilized) compounds (9–15a–c) and their analogues without lipophilization (9–15)”, shows the EC50 values only for anti-VZV, where are the results for HSV-1? For sample 12b, the EC50 is described as >200 µM, whereas for sample 12 is >10 µM. Why weren’t the exact values calculated? You mention that the EC50 were calculated for anti-VSV in the Methodology section, but the description is difficult to follow. Where are the results of TCID50 reduction for anti-VSV? Also, in “3.5.1. Viruses and cells” you mention that VERO cells were used for HSV-1 propagation, but there is no information on the cell line used for VSV culturing. Later, in “3.5.3. VSV inhibition analysis”, you wrote that “HEK-293T cells were infected with VSV Indiana strain”, however, the origin of HEK-293T is missing. Please include this information in “3.5.1. Viruses and cells”. In the “3.5.2. Cytotoxicity Assay”, the cytotoxicity was tested only towards VERO cells. Please include the results of cytotoxicity towards HEK-293T, which is a human-originating cell line, and this would provide valuable information. Moreover, the assessment of cytotoxicity towards HEK-293T is necessary to evaluate the results of anti-VSV activity.

In the description of cytotoxicity testing, you wrote that: “Most of the tested alkylated perylene compounds (at 10 µM) were not cytotoxic for Vero cells after a 48-h incubation (Figure 6). Compounds 10a, 12a and 14b decreased cell viability to ~80% compared with controls.”, whereas later, in the description of antiviral activity: “Regarding nucleoside-based alkylated perylene compounds, 9-butyl-substituted perylenylethynyl nucleosides 11a, 9a and 10a showed strong anti-HSV-1 activity and completely inhibited virus replication in Vero cells at 50 µM”. Please explain why have you used 50 µM in antiviral assays when the cytotoxicity clearly shows that at 10 µM the viability of VERO cells was reduced by 20% for 10a, 12a and 14b? There may be a mistake herein since, in Figure 6B, the concentration axis is labelled “Compounds [10 mM]”. Please correct. Why have you not calculated the EC50 for anti-HSV activity? The Methodology of antiviral assay needs a more detailed explanation.

You mention a photosensitization step in anti-VSV methodology, but the description needs clarification. Why there are no photosensitization steps in ant-HSV methods?

Please include a reference antiviral drug in your assays, especially for anti-HSV tests.

Comments on the Quality of English Language

The language is of an acceptable quality.

Reviewer 2 Report

Comments and Suggestions for Authors

The manuscript describes the synthesis of alkyl derivatives of perylene, their characterisation, spectroscopic properties, singlet oxygen production, molecular dynamics studies, and their antiviral activity in vitro against herpes simplex virus-1 and vesicular stomatitis virus. The antiviral activity of these new compounds was compared to non-alkylated perylene derivatives prepared by the authors in their previous work. Although it was ultimately shown that the described increase in lipophilicity of the compounds significantly reduced their antiviral activity, the work may be of interest to specialists in the field of medicinal chemistry.

The manuscript is properly organised and well written, with mostly good discussion and sound conclusions. Syntheses are mostly correctly described with appropriate compounds’ characterization. However, it is customary to write the quantities of products obtained, not only their yields. Furthermore, although I agree with the authors that the attachment of the alkyl chain at position 9 or 10 does not significantly affect the properties of the obtained compounds, given that mixtures of isomers were obtained as suggested, it would be good to estimate the ratio of individual isomers, if this is possible from their 1H NMR spectra. It would be good to add 1H NMR in the Supplementary materials, at least for all the final products, and check their purity on HPLC. Since perylene are known primarily as fluorophores, and since it is stated in the abstract that the obtained compounds were characterized as such, it would be useful to measure their fluorescence quantum yields.

Finally, the main weakness of the manuscript is the presentation of experiments with light. In the Materials and methods section, the measurements of the quantum yield of singlet oxygen are very narrowly described, and citing references is not enough considering that in this work different oxygen trap was used, solvent is unknown, and it would be useful to add light specification/irradiation parameters here, and in Table 1. Irradiation conditions are completely unknown (what kind of light source was used, wavelength, dose of light etc.) for all presented biological studies, so this should be addressed in the experimental part, Table 2 and Figure 6, as well as in the discussion.

Reviewer 3 Report

Comments and Suggestions for Authors

Mikhnovets et al. in their manuscript "Alkyl derivatives of perylene photosensitizing antivirals..." describe the synthesis of new compounds with potential antiviral applications. The authors introduced a series of modifications to the aromatic core of perylene-based compounds and investigated the antiviral properties on HSV-1 and VSV. The manuscript is written in good English and is relatively easy to read.

The focus of the manuscript is on the synthesis of the compounds and a very basic characterization. The biological tests are limited to viability studies and reduction of infectivity.

Although the experiments are performed at a high level, the main weakness of the manuscript is the limited novelty and limited biological tests that would lead to particualr conclusions. There are large number of similar synthesis and compounds reported. The experiments presented do not show that the compounds tested intercalate into the viral envelope and reduce infectivity, i.e., based on the present experiments, the molecular mechanisms cannot be confirmed. The authors have demonstrated reduced replication of VSV and HSV-1 after treatment with the compounds, but they do not provide evidence of membrane damage, reduced infectivity (e.g., DNA analysis), etc. The manuscript also does not indicate whether these photosensitizers should be treated with light. The actual molecular mechanism and application is not event sufficiently discussed.

Minor:

1.            Spell out viruses in the first sentence

2.            There is a VZV vaccine (which has been used successfully for several years)

3.            Resistance is not a reason why nucleoside analogs do not work for all viruses, but rather specificity for the compound

4.            Please provide references for VSV in periodontal disease

5.            Supercapsid is an uncommon term, envelope is sufficient

6.            Figure 1. is nice, but it is not clear how it relates to the manuscript and the compounds studied

Round 2

Reviewer 1 Report

Comments and Suggestions for Authors

Dear Authors,

Thank you very much for responding to my previous comments. The manuscript has been significantly improved.

Comments on the Quality of English Language

This text is written with acceptable English quality, with minor spelling, grammar, and punctuation errors.

Author Response

No changes requested by Reviewer.

Reviewer 2 Report

Comments and Suggestions for Authors

The authors addressed most of the questions raised in the review, and the biological part of the manuscript was significantly expanded. However, it is still unclear under what conditions the singlet oxygen quantum yield was measured, which is quite important for this work.

For photodynamic inactivation studies, two controls are usually performed and required - a dark control and a light-only control. Here, daylight was used as a control instead of a dark control, and there appears to be no light-only control. The authors need to clarify this.

The text below Figure 6 mentions compound 11 c and refers to Figure 6B, however 11 c is missing from Figure 6A and Figure 6B.

Author Response

The authors addressed most of the questions raised in the review, and the biological part of the manuscript was significantly expanded. However, it is still unclear under what conditions the singlet oxygen quantum yield was measured, which is quite important for this work.

Conditions for singlet oxygen measurements has been added (section 3.3).

For photodynamic inactivation studies, two controls are usually performed and required - a dark control and a light-only control. Here, daylight was used as a control instead of a dark control, and there appears to be no light-only control. The authors need to clarify this.

We thank the reviewer for his comment. Non-alkylated perylene compounds were used as controls in experiments (1) in daylight and (2) under blue light irradiation. We did NOT used non-alkylated perylene compounds as controls in experiments performed under red light (in the absence of excitation light/daylight), because the antiviral activity of non-alkylated (control) compounds under red light was studied in detail in our previous work [23,24] using several enveloped viruses; these compounds lost their antiviral activity under red light, similar to their alkylated counterparts. This is explained in the main text, see the sentence: “Similar to other recently described perylene compounds [21,23,24], the alkylated perylene derivatives lost their HSV-1-inactivating activity when the entire experiment....” In addition, acyclovir was used as another control in the daylight-based experiments because acyclovir is a well-established anti-HSV-1 nucleoside analogue. However, acyclovir was NOT used in studies on the light-dependent antiviral activity of the investigated compounds, as its mechanism of action differs significantly from that of the photosensitizing antivirals (termination of DNA replication).

The text below Figure 6 mentions compound 11 c and refers to Figure 6B, however 11 c is missing from Figure 6A and Figure 6B.

Corrected. 11c was not tested in anti-HSV-1 assays.

Reviewer 3 Report

Comments and Suggestions for Authors

The authors have responded appropriately to all my comments and have added a large amount of new data that has significantly increased the value of the manuscript.

Minor: the conclusion that the compound prevent fusion could be premature as the authors do not specify DNA levels after entry, i.e. it is possible that other mechanisms at the DNA level, intracellular transport, etc. may also reduce replication.

There are some minor errors and typos that should be carefully corrected at the proof level.

Author Response

The authors have responded appropriately to all my comments and have added a large amount of new data that has significantly increased the value of the manuscript.
Minor: the conclusion that the compound prevent fusion could be premature as the authors do not specify DNA levels after entry, i.e. it is possible that other mechanisms at the DNA level, intracellular transport, etc. may also reduce replication.
The mechanism of action of perylene-based compounds has been extensively studied in the past (e.g. refs 19-25), and the results indicate that these compounds inhibit the formation of negative membrane curvature during virus-cell fusion, which eventually leads to the blockage of the virus-cell fusion process. Based on the structural similarities between the previously described “old” compounds [19-25] and the “new” derivatives used in this work, we can hypothesise the same mechanism of their antiviral activity based on (1) intercalation into the viral envelope, (2) oxidation of the envelope lipids by singlet oxygen generated after blue light irradiation, (3) blocking the negative membrane curvature, and finally (4) blocking the virus-cell fusion machinery. (discussed also in the main text).